# Phage-induced efflux down-regulation boosts antibiotic efficacy

**Samuel Kraus[1], Megan L. Fletcher[1], Urszula Łapińska[1], Krina Chawla[1], Evan Baker[2,3], Erin L. Attrill[1], Paul O'Neill[4], Audrey Farbos[4], Aaron Jeffries[4], Edouard E. Galyov[5], Sunee Korbsrisate[6], Kay B. Barnes[7], Sarah V. Harding[7], Krasimira Tsaneva-Atanasova[2,3], Mark A. T. Blaskovich[8], Stefano Pagliara[1‡]***

**1** Living Systems Institute and Biosciences, University of Exeter, Exeter, Devon, United Kingdom,
**2** Department of Mathematics and Living Systems Institute, University of Exeter, Exeter, Devon, United Kingdom, **3** EPSRC Hub for Quantitative Modelling in Healthcare, University of Exeter, Exeter, United Kingdom, **4** Biosciences, University of Exeter, Exeter, Devon, EX4 4QD, United Kingdom, **5** Department of Genetics and Genome Biology, University of Leicester, University Road, Leicester, United Kingdom, **6** Department of Immunology, Faculty of Medicine Siriraj Hospital, Mahidol University, Bangkok Thailand, **7** Defence Science and Technology Laboratory, Porton Down, Salisbury, Wiltshire, United Kingdom, **8** Centre for Superbug Solutions, Institute for Molecular Bioscience, The University of Queensland, St. Lucia, Queensland, Australia

‡ Lead contact
* s.pagliara@exeter.ac.uk

**Data Availability Statement:** All data generated or analysed during this study are included in this published article and its supplementary information files.

## Abstract

The interactions between a virus and its host vary in space and time and are affected by the presence of molecules that alter the physiology of either the host or the virus. Determining the molecular mechanisms at the basis of these interactions is paramount for predicting the fate of bacterial and phage populations and for designing rational phage-antibiotic therapies. We study the interactions between stationary phase *Burkholderia thailandensis* and the phage ΦBp-AMP1. Although heterogeneous genetic resistance to phage rapidly emerges in *B. thailandensis*, the presence of phage enhances the efficacy of three major antibiotic classes, the quinolones, the beta-lactams and the tetracyclines, but antagonizes tetrahydrofolate synthesis inhibitors. We discovered that enhanced antibiotic efficacy is facilitated by reduced antibiotic efflux in the presence of phage. This new phage-antibiotic therapy allows for eradication of stationary phase bacteria, whilst requiring reduced antibiotic concentrations, which is crucial for treating infections in sites where it is difficult to achieve high antibiotic concentrations.

## Author summary

Bacteriophages are viruses that infect and kill bacteria and therefore are considered to be a very valuable alternative to the antibiotic molecules that are currently employed in the clinic but towards which bacteria are becoming increasingly resistant. Here we show that when infected with a bacteriophage termed ΦBp-AMP1, the bacterium *Burkholderia thailandensis* becomes more susceptible to three major antibiotic classes routinely employed to treat infections caused by *Burkholderia*. We discovered that enhanced antibiotic

**Funding:** This work was supported by the BBSRC through a grant awarded to S.P., K.T.A. and U.L. (BB/V008021/1) and the EPSRC through a grant awarded to S.P. (EP/Y023528/1). KTA and E.B. gratefully acknowledge the financial support of the EPSRC (EP/T017856/1). This project utilised equipment funded by a Wellcome Trust Institutional Strategic Support Fund (WT097835MF), a Wellcome Trust Multi-User Equipment Award (WT101650MA) and a BBSRC LoLa award (BB/K003240/1). The funders had no role in study design, data collection and analysis, decision to publish, or preparation of the manuscript. For the purpose of open access, the authors have applied a 'Creative Commons Attribution (CC BY) licence to any Author Accepted Manuscript version arising from this submission.

**Competing interests:** The authors have declared that no competing interests exist.

efficacy in the presence of this bacteriophage is due to a decreased capability of the bacterium to expel antibiotics when it is infected with this bacteriophage. Understanding how bacteriophages and antibiotic molecules interact with each other is essential for optimizing bacteriophage-antibiotic therapy against bacterial infections.

## Introduction

Antimicrobial resistance (AMR) has a dramatic impact on global health with an estimated 5 million deaths associated with AMR in 2019 alone [1]. *Burkholderia (B.) pseudomallei*, *B. thailandensis* and *B. mallei* (known as the Bptm group) cause life-threatening diseases [2–5]. Of particular interest is *Burkholderia pseudomallei* which is endemic in at least 79 tropical and subtropical countries and causes the life-threatening disease melioidosis [6]. Case numbers of melioidosis are closely linked to extreme weather events with heavy rainfall and environmental disturbance, such as landslides or construction work, exacerbating outbreaks [6]. In northern Australia, First Nations people are often disproportionally impacted, likely due to comparatively higher numbers of people living in regions with endemic melioidosis, as well as often poor access to state-of-the-art treatment [7].

Bptm group members are challenging to treat with currently available antibiotics as they are intrinsically resistant to aminoglycosides, macrolides and oxazolidinones due to constitutively expressed efflux pumps [8–11], whereas an atypical lipopolysaccharide structure plays a crucial role in resistance to cationic peptides such as polymyxins [2]. Moreover, misuse and overuse of antibiotics in the food industry, animal husbandry, and medicine, as well as environmental dispersion of antibiotics in ground water through livestock excretions and increased endemic bacterial spread due to global warming, have recently contributed to the spread of acquired genetic resistance [12]. Bptm species can acquire antibiotic resistance *in vivo* during treatment, which can be fatal if treatment is not shifted to alternative drugs in due course [2].

In order to overcome the development of resistance to antibiotic monotherapies, the deployment of two or more different antibiotics in combination therapies has shown some success in treating and preventing infections [13,14]; however, antibiotics in conventional regimens can antagonize each other [15,16]. Therefore, several alternatives to traditional antibiotics have been explored in order to treat resistant infections, including: antibody therapy, antimicrobial peptides, probiotics, metal chelation, CRISPR-Cas9, bioengineered toxins, bacteriocins, and vaccines [17–22]. Although promising, all of these approaches are limited by the fact that these antimicrobial agents cannot evolve to overcome bacterial resistance, a limitation shared with antibiotic molecules routinely employed in healthcare [23].

In contrast, like bacteria, bacteriophages (i.e., viruses that infect bacteria) amplify at the site of an infection and evolve to overcome bacterial resistance; therefore, also due to their high number and genetic variability, phage constitute a large and valuable reservoir of natural antimicrobials [24]. However, the use of phage as a monotherapy to treat bacterial infections presents several challenges, particularly narrow host range and the rapid evolution of resistance to phage [25]. As antibiotics are the current standard of care, using phage as an adjuvant to antibiotics instead of a monotherapy may be a more rational therapeutic use of phage [26]. Therefore, phage-antibiotic therapy has undergone a robust revitalization in recent years [27–29].

Enhanced efficacy of combination therapies has historically been attributed to an increase in phage produced from bacteria in the presence of β-lactam antibiotics, relative to production in their absence [30,31]. More recently, a variety of lytic phage was observed to form larger

plaques in the presence of sublethal concentrations of β-lactam, quinolone and tetracycline antibiotics [32–35]. This effect has been termed phage-antibiotic synergy and has been linked to antibiotic-induced bacterial filamentation which accelerates phage assembly and cell lysis [32]. However, other recent evidence suggests that phage-antibiotic synergy can be obtained independently of cell filamentation, enhancement of phage production or the strict use of lytic phage, and that temperate phage are also viable for phage-antibiotic therapy if prophages are induced by DNA damaging antibiotics [36]. Furthermore, the evolution of bacterial resistance to phage that use efflux pumps as receptors can alter the efflux pump mechanism, causing increased sensitivity to antibiotics [37].

Phage-antibiotic synergy has mostly been studied with only one or two concentrations of the antimicrobials, which is insufficient to predict combinatorial concentrations that are effective during treatment, leading to mixed results during phage-antibiotic therapy investigations [26,38]. Indeed, a recent study employing *Escherichia coli* and the lytic phage φHP3 tested several orders of magnitude of both phage and antibiotic concentrations, finding that the nature of the interaction between antibiotics and phage depended both on the type and concentration of the antibiotic and phage employed [39].

Therefore, it is imperative to discover and understand the molecular mechanisms at the basis of the interactions between phage and antibiotics, in order to rationally design successful new phage-antibiotic therapy [24]. Here we test molecules representative of eight major antibiotic classes in combination with a recently discovered phage, termed ΦBp-AMP1, that infects and kills *Burkholderia pseudomallei* and *Burkholderia thailandensis* [40–42]. ΦBp-AMP1 is a podovirus with a 45 nm icosahedral capsid, a 20 nm non-contractile tail and a 45 kb genome. In contrast to other *Burkholderia* phage that are strictly lytic [43], ΦBp-AMP1 displays a temperature-dependent switch from the temperate (at 25˚C) to the lytic cycle (at 37˚C) [40–42]. Using optically based microtiter plate assays and genomics, we studied the development of genetic resistance to ΦBp-AMP1 in stationary phase *B. thailandensis*. We investigated the dynamics of the interactions between ΦBp-AMP1 and representative molecules of the major antibiotic classes over multiple orders of magnitude of antibiotic concentrations and phage titers. We determined the molecular mechanisms that support these phage-antibiotic interactions via single-cell microscopy, mathematical modelling and gene expression profiling of stationary phase *B. thailandensis* undergoing phage-antibiotic combination therapy as well as antibiotic or phage monotherapy.

## Results

### Genetic resistance to ΦBp-AMP1 phage in *B. thailandensis* is heterogeneous

We used well-mixed 50 mL liquid cultures of stationary phase *B. thailandensis* (strain E264) with ΦBp-AMP1 at a multiplicity of infection (MOI) of 1 in lysogeny broth (LB) medium at 37˚C and performed colony forming unit (CFU) assays every 2 h over a 24 h period. For the first 14 h of incubation in the presence of phage, we did not observe neither a reduction nor an increase of the bacterial population, despite the presence of LB medium allowed regrowth of uninfected stationary phase *B. thailandensis* cultures (Fig 1A and Data A in S1 File).

The phage population instead increased within the first 4 h of co-incubation and reached a plateau after 6 h (S1 Fig and S2 File). The bacterial population started to increase after 14 h co-incubation with phage and after 24 h reached a plateau 2-fold lower compared to that measured in the absence of phage (Fig 1A). These data suggest that a bacterial subpopulation survives phage treatment, passes on phage immunity to its progeny and becomes the predominant genotype within the population; in contrast, a susceptible subpopulation

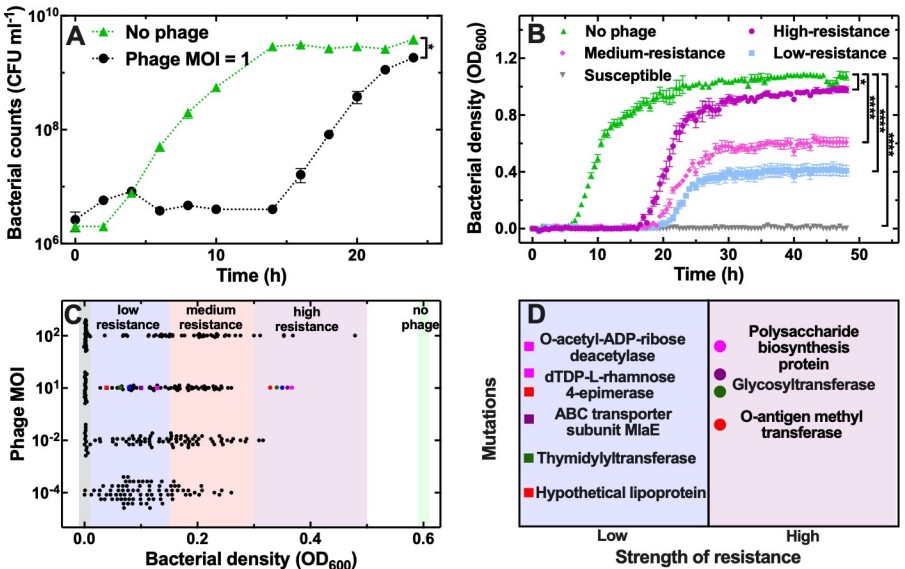

**Fig 1. Heterogenous resistance to ΦBp-AMP1 in *B. thailandensis*.** (A) Regrowth of stationary phase *B. thailandensis* populations in the presence of LB medium only (green triangles) or together with phage at an MOI of 1 (black circles). Symbols and error bars are means and standard errors of the means of CFU measurements obtained from biological triplicates each containing technical triplicates. Very small error bars cannot be visualised due to overlap with the datapoints. Dotted lines are guides-for-the-eye. Numerical values are provided in Data A in S1 File. Corresponding phage counts are reported in S1 Fig. (B) Regrowth of stationary phase *B. thailandensis* populations in the presence of LB medium only ($1.05 < OD_{600} < 1.2$ after 48 h, green triangles) or together with phage at an MOI of 1 with different levels of bacterial resistance to phage emerging: high-resistance ($0.9 < OD_{600} < 1.05$ after 48 h, purple circles), medium-resistance ($0.5 < OD_{600} < 0.7$ after 48 h, magenta diamonds), low-resistance ($0.3 < OD_{600} < 0.5$ after 48 h, blue squares), susceptible ($0 < OD_{600} < 0.01$ after 48 h, grey downward triangles). Symbols and error bars are means and standard errors of bacterial density values, measured in $OD_{600}$, obtained from 84 technical replicates from biological triplicates. Very small error bars cannot be visualised due to overlap with the datapoints. * indicate a p-value $< 0.05$, **** indicate a p-value $< 0.0001$. Numerical values are provided in Data B in S1 File. (C). Bacterial density measurements after 24 h in the presence of phage at an MOI of $10^{-4}$, $10^{-2}$, 1 and $10^2$. Each black circle represents a bacterial density value performed on one of 84 technical micro-culture replicates from biological triplicates. The coloured vertical bands in the background indicate the range of $OD_{600}$ values for each level of resistance after 24 h: high-resistance ($0.3 < OD_{600} < 0.5$, purple band), medium-resistance ($0.15 < OD_{600} < 0.3$, magenta band), low-resistance ($0.01 < OD_{600} < 0.15$, blue band), susceptibility ($0 < OD_{600} < 0.01$, grey band). Numerical values are provided in Data C in S1 File. Coloured circle and square data points indicate populations that were whole genome sequenced; the corresponding unique mutations identified for each population are reported in (D).

becomes a minority due to phage infection but persists in the presence of phage since the phage population does not decline within our experimental time frame (S1 Fig).

To test this hypothesis, we infected 84 stationary phase bacterial micro-cultures in 100 μL well plates and measured the bacterial density in the presence of phage and LB medium. We found that some bacterial micro-cultures contained only bacteria that were susceptible to phage and did not grow; whereas in other bacterial micro-cultures genetic resistance to phage emerged and the cultures started to regrow in the presence of LB medium, albeit with different onset times, slopes and saturations of growth (Fig 1B and Data B in S1 File). Overall, the distribution of growth levels observed in micro-cultures could be described by a hurdle-gamma distribution (Figs 1C, S2 and Data C in S1 File), where the hurdle describes the cultures that are susceptible to phage, whereas the gamma describes the spread of growth in the cultures that are resistant to phage. These data suggest that *B. thailandensis* populations must contain a heterogeneous pool of mutants that are resistant to the phage. In larger scale cultures we did not observe such heterogeneity (Fig 1A) suggesting that it is critical to reduce the bacterial

inoculum (from $10^8$ to $10^5$ bacteria) to capture a heterogeneous pool of susceptible and resistant populations.

Next, we set out to investigate whether the phage MOI has an impact on the emergence of a heterogeneous pool of genetically resistant mutants. We infected 84 stationary phase bacterial micro-cultures with four different phage MOIs and measured the bacterial density after 24 h. We found that the proportion of cultures susceptible to phage increased with the MOI; the fraction of cultures displaying low resistance to phage or high resistance to phage reduced and increased, respectively, with increasing phage MOI (Figs 1C, S2 and Data C in S1 File). Moreover, all of the cultures grew when treated with ΦBp-AMP1 at an MOI of 1 at 25˚C (S3 Fig) and Data C in S1 File), confirming that ΦBp-AMP1 features a temperature-dependent switch from the lytic to the lysogenic cycle.

Next, we harvested five survivor low-resistance and five survivor high-resistance populations (indicated with colored squares and circles, respectively, in Fig 1C) and re-inoculated them into 100 μL well plates in the presence of phage for 48 h. We found that the high-resistance populations started growth earlier and reached higher final densities compared to low-resistance populations (S4A and S4B Fig, respectively and S3 File). As expected, we found a positive correlation between the final optical density at the end of this second 48h exposure to phage and the optical density measured for each survivor population at the end of the first 48h exposure to phage (r = 0.81, **, S4C Fig) and a negative correlation between the onset of growth during this second exposure to phage and the optical density measured for each survivor population at the end of the first 48h exposure to phage (r = -0.69, *, S4D Fig).

To investigate the mechanisms that enable resistance to phage in these ten phage-resistant populations we performed whole genome sequencing on each population. We found that these phage-resistant populations harboured unique mutations (i.e. the only genetic changes observed when compared to the parental genome) in genes encoding the following products (in the following order from the highest to the lowest resistant population sampled from Fig 1C): a polysaccharide biosynthesis protein, a glycosyltransferase, the O-antigen methyl transferase, the O-acetyl-ADP-ribose deacetylase, the dTDP-L-rhamnose-4-epimerase, the ABC transporter subunit MlaE, the thymidylyltransferase and a hypothetical lipoprotein (Fig 1D and S1 Table). Moreover, we did not find lysogenic ΦBp-AMP1 in any of the genomes sequenced (S2 Table), suggesting that stable lysogeny did not occur at 37˚C. We found two common temperate bacteriophages of *Burkholderia* species, ΦE12-2 and ΦE125, in all sequenced *B. thailandensis* genomes (S2 Table).

Taken together, these data suggest that ΦBp-AMP1 is a lytic phage of *B. thailandensis* at 37˚C, that different levels of genetic resistance can emerge within putatively clonal populations of *B. thailandensis* depending on the phage MOI employed and that mutations of genes encoding membrane associated proteins confer high resistance to ΦBp-AMP1. Therefore, in order to make ΦBp-AMP1 effective in eradicating stationary phase *B. thailandensis* there is a need to use this phage in combination with clinically relevant antibiotics.

## ΦBp-AMP1 phage increases the growth inhibitory efficacy of quinolones, β-lactams and tetracyclines

We set out to investigate whether the use of ΦBp-AMP1 increases the efficacy of clinically relevant antibiotics in inhibiting the regrowth of stationary phase *B. thailandensis* when incubated in LB medium. We used four major antibiotic classes commonly employed for treatment of melioidosis, namely quinolones, β-lactams, tetracyclines and tetrahydrofolate synthesis inhibitors as well as antimicrobial agents that are not routinely employed to treat melioidosis, namely aminoglycosides, oxazolidinones, macrolides and glycopeptides (Table 1). We

**Table 1. ΦBp-AMP1 phage increases the inhibitory efficacy of quinolones, β-lactams and tetracyclines.** Antibiotic class, antibiotic molecule, minimum inhibitory concentration (MIC) measured against stationary phase *B. thailandensis* when used as monotherapy or in combination therapy with phage ΦBp-AMP1 at an MOI of 1 and the fractional inhibitory concentration (FIC) as the ratio of the MIC value obtained from the combination therapy over the MIC value obtained from monotherapy. The initial bacterial inoculum was $5 \times 10^5$ cells mL$^{-1}$, regrowth of stationary phase bacteria was measured as optical density after 24 h of monotherapy or combination therapy compared with regrowth of untreated stationary phase bacteria incubated in LB medium only. The MIC value after 24 h was determined as the antibiotic concentration value for which the bacterial growth was less than 10% the growth value measured for untreated bacteria. Each measurement was performed in biological triplicates each consisting of five technical replicates.

| Class | Molecule | Monotherapy MIC (µg ml$^{-1}$) | Co-treatment MIC (µg ml$^{-1}$) | FIC |
|---|---|---|---|---|
| Quinolones | Nalidixic Acid | 16 | 4 | 0.25 |
| | Ciprofloxacin | 2 | 0.125 | 0.06 |
| | Ofloxacin | 4 | 0.5 | 0.125 |
| | Levofloxacin | 4 | 0.5 | 0.125 |
| | Finafloxacin | 2 | 0.06 | 0.03 |
| | Moxifloxacin | 2 | 0.25 | 0.125 |
| β-Lactams | Amoxicillin | 64 | 16 | 0.25 |
| | Ampicillin | 64 | 16 | 0.25 |
| | Cefaclor | 256 | 64 | 0.25 |
| | Ceftazidime | 2 | 0.25 | 0.125 |
| | Meropenem | 8 | 2 | 0.25 |
| Tetracyclines | Doxycycline | 4 | 1 | 0.25 |
| | Tetracycline | 8 | 2 | 0.25 |
| Tetrahydrofolate Synthesis Inhibitors | Trimethoprim (TMP) | 32 | 32 | 1 |
| | Sulfamethoxazole (SMX) | 1024 | 1024 | 1 |
| | Co-trimoxazole (TMP/SMX) | 32/160 | 32/160 | 1 |
| Aminoglycosides | Gentamicin | >512 | >512 | - |
| | Streptomycin | >512 | >512 | - |
| Oxazolidinones | Linezolid | >512 | >512 | - |
| Macrolides | Roxithromycin | >512 | >512 | - |
| Glycopeptides | Vancomycin | >512 | >512 | - |

determined the fractional inhibitory concentration (FIC) index of each antibiotic against stationary phase *B. thailandensis* by dividing the minimum inhibitory concentration (MIC) derived from combination therapy with phage at an MOI of 1 by the MIC of antibiotic monotherapy. Therefore, the smaller the FIC index value measured, the higher is the increase in antibiotic efficacy of the combination therapy.

Remarkably, combination therapy increased the efficacy of a diverse range of molecules representative of all of the fluoroquinolone generations. Specifically, we measured an FIC index of 0.25 for nalidixic acid (1st generation fluoroquinolone), 0.06 for ciprofloxacin (2nd generation), 0.125 for levofloxacin (3rd generation), 0.125 for moxifloxacin (4th generation) and 0.03 for finafloxacin (5th generation). Similarly, combination therapy increased the efficacy of representative β-lactams and tetracyclines against *B. thailandensis*; however, combination therapy did not increase the efficacy of tetrahydrofolate synthesis inhibitors (Table 1).

Inspired by these successful combination therapy findings, we set out to determine whether combination therapy with phage increased the efficacy of antibiotics that are not routinely employed to treat melioidosis. However, combination therapy with phage did not increase the efficacy of two aminoglycosides, an oxazolidinone, a macrolide or a glycopeptide (Table 1), antibiotics against which *B. thailandensis* is intrinsically resistant due to constitutively expressed efflux pumps and an atypical lipopolysaccharide structure [2].

Next, we investigated the bacterial population dynamics in the presence of phage and a sub-inhibitory concentration of levofloxacin, cefaclor or trimethoprim for which combination therapy with phage provided high, medium or no increase in efficacy, respectively (Table 1). Combination therapy with 0.25× MIC levofloxacin and phage at an MOI of 1 in LB medium did not allow for regrowth of stationary phase *B. thailandensis* within the 24 h experimental timeframe, whereas regrowth started after 5 h exposure to levofloxacin monotherapy and 16 h after exposure to phage monotherapy (Fig 2A and Data A in S4 File).

Following cefaclor monotherapy at 0.25× its MIC, the stationary phase *B. thailandensis* population started to regrow after 5 h, whereas following either phage monotherapy at an MOI of 1 or combination therapy with 0.25× MIC cefaclor and phage at an MOI of 1, the stationary phase *B. thailandensis* population started to regrow after 16 h and reached a plateau that was significantly lower compared to cefaclor monotherapy (Fig 2B and Data B in S4 File). Following trimethoprim monotherapy at 0.25× its MIC, the stationary phase *B. thailandensis* population started to regrow after 9 h. Similarly, following combination therapy with 0.25× MIC trimethoprim and phage at an MOI of 1, the stationary phase *B. thailandensis* population started to regrow after 9 h and after 24 h reached a plateau that was significantly higher compared to that reached during phage monotherapy (red triangles and black circles in Fig 2C, respectively, and Data C in S4 File).

Taken together these data demonstrate that combination therapy with phage ΦBp-AMP1 increases the inhibitory efficacy of quinolones, β-lactams and tetracyclines, whilst combination therapy with tetrahydrofolate synthesis inhibitors decreases the inhibitory efficacy of ΦBp-AMP1.

## The interactions between phage ΦBp-AMP1 and antibiotics depend on the antibiotic but not on the phage concentration

Next, we set out to understand whether phage-antibiotic interactions in inhibiting bacterial growth depend on either the antibiotic or the phage concentration. Firstly, we measured the bacterial density of stationary phase *B. thailandensis* over time during exposure to LB medium and different antibiotic concentrations at a constant phage MOI of 1. In order to estimate bacterial density and the associated uncertainty also for antibiotic concentrations that we did not investigate experimentally, we fitted these data to a statistical non-linear regression model. This model also allowed us to infer the probability of additivism between the phage effect and the antibiotic effect on bacterial growth. We summarised our results in the form of interaction plots [39] reporting bacterial density values in the absence of phage or antibiotic, growth values

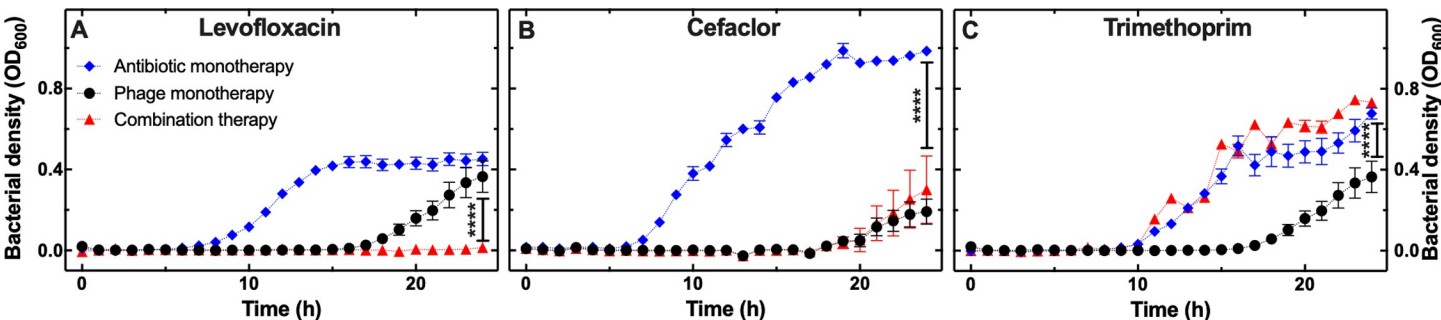

**Fig 2. Bacterial population dynamics in the presence of ΦBp-AMP1 and sub-inhibitory antibiotic concentrations.** Regrowth of stationary phase *B. thailandensis* populations in the presence of LB medium and either phage at an MOI of 1 (black circles), or 0.25× MIC of (A) levofloxacin, (B) cefaclor or (C) trimethoprim either as monotherapy (blue diamonds) or in combination therapy with phage at an MOI of 1 (red triangles). Symbols and error bars are means and standard errors of the means of bacterial density values, measured in OD$_{600}$, obtained from biological triplicates each consisting of five technical replicates. Very small error bars cannot be visualised due to overlap with the datapoints. Dotted lines are guides-for-the-eye. **** indicate a p-value < 0.0001. Numerical values are reported in S4 File.

following phage or antibiotic monotherapies and growth values following combination therapy.

When ciprofloxacin was used above its MIC, we measured growth suppression following both monotherapy and combination therapy (Fig 3A and Data A in S5 File), without additive effect (blue shaded area in Fig 3B and 3E) in line with previous reports [39]. In contrast, when ciprofloxacin was used at sub-inhibitory concentrations, the model predicted an interacting region: successful growth suppression was achieved only in the case of combination therapy (red lines in Fig 3A), with a degree of confidence of >95% in predicted additive effect at ciprofloxacin concentrations between 0.06 and 2 µg ml$^{-1}$ (red shaded area in Fig 3B and 3D). This additive concentration range was broad for ciprofloxacin, finafloxacin and moxifloxacin and narrower for nalidixic acid, ofloxacin and levofloxacin (S3 Table). Finally, we did not record an interaction effect between phage and ciprofloxacin at low antibiotic concentrations (white shaded area in Fig 3B and 3C) in line with previous reports [39].

We recorded similar concentration-dependent interactions between phage and cefaclor (Fig 3F–3J and Data A in S5 File); however, for this antibiotic the additive effect was limited to the antibiotic range 64–256 µg mL$^{-1}$, i.e. a 4-fold additive concentration range (Fig 3G). In

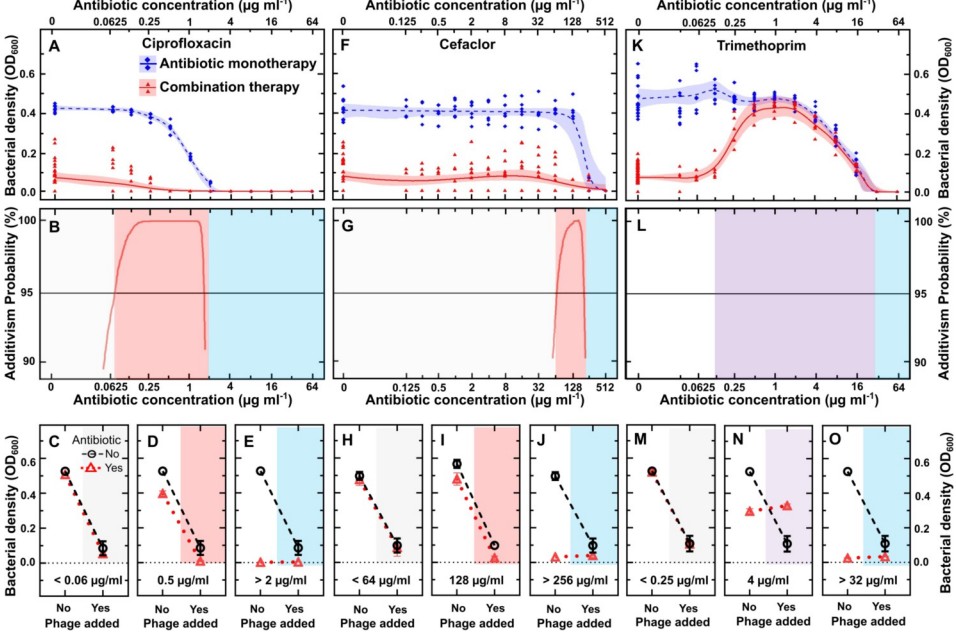

**Fig 3. Interactions between ΦBp-AMP1 and antibiotics depend on the antibiotic concentration.** Experimental data (symbols) and model predictions (lines and bands) describing the dependence of bacterial density on the concentration of (A) ciprofloxacin, (F) cefaclor or (K) trimethoprim in the absence (blue diamonds) and presence of phage ΦBp-AMP1 at an MOI of 1 (red triangles) after 24 h treatment. Each symbol represents the bacterial density measured in one of 15 technical replicates collated from biological triplicates. Some of the symbols overlap with each other. The lines and shaded areas are the medians, upper and lower quartiles, estimated by fitting our statistical non-linear regression model to our experimental data via Markov Chain Monte Carlo simulations. Corresponding predicted probability of an additive interaction between phage and (B) ciprofloxacin, (G) cefaclor and (L) trimethoprim (red lines) is shown only for antibiotic concentration ranges where the probability is higher than 90% (red shaded areas). Not shaded or blue shaded areas indicate antibiotic concentration ranges where the phage or the antibiotic dominate, respectively. Purple shaded areas indicate antagonism. Corresponding interaction plots at antibiotic concentrations selected from the ranges above for (C-E) ciprofloxacin, (H-J) cefaclor and (M-O) trimethoprim. Black circles connected by dashed lines show bacterial density values following control experiments and phage monotherapy, red triangles connected by dotted lines show bacterial density values following antibiotic monotherapy and combination therapy. Numerical values are reported in Data A S5 File.

addition, ceftazidime, meropenem, doxycycline, tetracycline, amoxicillin and ampicillin displayed a narrow additive concentration range (S3 Table).

In contrast, we found an antagonistic effect when phage was used in combination with trimethoprim concentrations in the range of 0.125–32 μg mL$^{-1}$ with the line connecting the growth value following antibiotic therapy and combination therapy (red, dotted line in Fig 3N and Data A in S5 File) having a less negative slope than the line connecting the growth value following control experiments and phage monotherapy (black, dashed line in Fig 3N). A similarly extended antagonistic range was recorded when phage was used in combination with either sulfamethoxazole or co-trimoxazole (S5 Fig and Data B in S5 File).

Secondly, we measured the bacterial density of stationary phase *B. thailandensis* over time during incubation in LB medium while simultaneously varying the initial antibiotic concentration and phage MOI. We used ciprofloxacin, ampicillin and trimethoprim as representative molecules displaying a broad additive concentration range, a narrow additive concentration range and antagonism, respectively, in combination with phage at an initial MOI of either $10^{-4}$, $10^{-2}$, $10^{0}$ or $10^{2}$. We found that growth inhibition was significantly extended below the MIC of ciprofloxacin or ampicillin even at a phage MOI of $10^{-4}$ (Fig 4A, 4B and S6 File), suggesting that a modest phage concentration is sufficient for increasing ciprofloxacin and ampicillin inhibitory efficacy against *B. thailandensis*.

In contrast, growth inhibition was not extended below the MIC of trimethoprim at any of the phage MOIs tested (Fig 4C and S6 File). Moreover, in the presence of phage *B. thailandensis* growth was maximal for trimethoprim concentrations in the range 1–8 μg mL$^{-1}$ and decreased at lower trimethoprim concentrations (Fig 4C). Therefore, these data confirm the hypothesis above that sub-inhibitory concentrations of trimethoprim antagonize with phage efficacy to inhibit bacterial growth. Taken together, these data demonstrate that the interaction between phage ΦBp-AMP1 and antibiotics strongly depends on the antibiotic concentration and mode of action but not on the initial phage MOI.

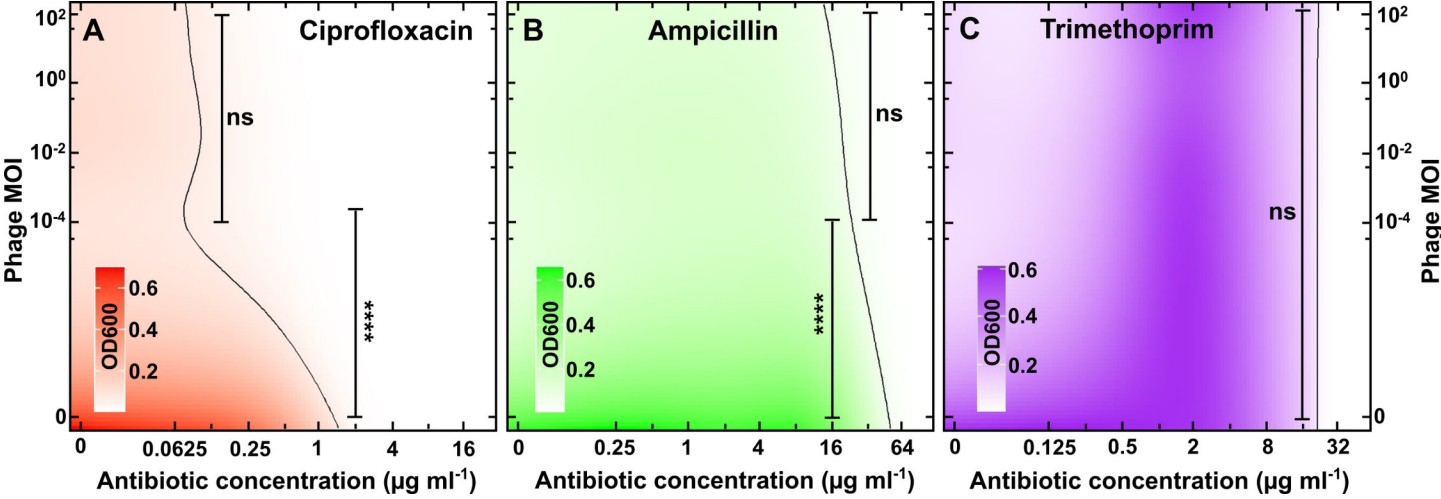

**Fig 4. Phage-antibiotic interactions are not affected by the phage concentration.** Heatmaps of *B. thailandensis* density (measured in OD$_{600}$) after 24 h treatment with different initial phage MOIs and different concentrations of (A) ciprofloxacin, (B) ampicillin or (C) trimethoprim. Heatmaps were obtained via hierarchical Bayesian statistical modelling fitted to our experimental data (measured only at the phage and antibiotic concentrations indicated on the x- and y-axes). The vertical black lines are predictions of all antibiotic-phage combinations that permit bacterial density values that are lower than 10% of the bacterial density values obtained for bacteria growing in LB medium only. ns indicates no statistical significance, **** indicates a p-value < 0.0001. Numerical values are reported in S6 File.

## Synergistic bactericidal effect of phage and ciprofloxacin in combination therapy

We next set out to quantify the bactericidal efficacy of phage-antibiotic combination therapy. We exposed stationary phase *B. thailandensis* to a range of concentrations of ciprofloxacin and phage at an MOI of 1 for 24 h, we quantified bactericidal efficacy by measuring survivors via CFU assays and calculated the survivor fold reduction compared to untreated bacteria (i.e. the higher the fold reduction in Fig 5A the stronger the bactericidal effect). Combined with phage, sub-MIC concentrations of ciprofloxacin allowed for a survivor fold reduction that was between 3 and 120 times greater compared to ciprofloxacin monotherapy (red and blue bars in Fig 5A, respectively, p-value < 0.0001 for all pair-wise comparisons); thus, suggesting a synergistic bactericidal effect between phage and ciprofloxacin, considering that phage monotherapy provided only a 2-fold survivor reduction (Fig 5A and Data A in S7 File). Moreover, at supra-MIC concentrations, combination therapy achieved a survivor fold reduction that was between 570 and 2400 times greater compared to ciprofloxacin monotherapy (p-value < 0.0001 for all pair-wise comparisons a part from 8× MIC). Notably, the minimum

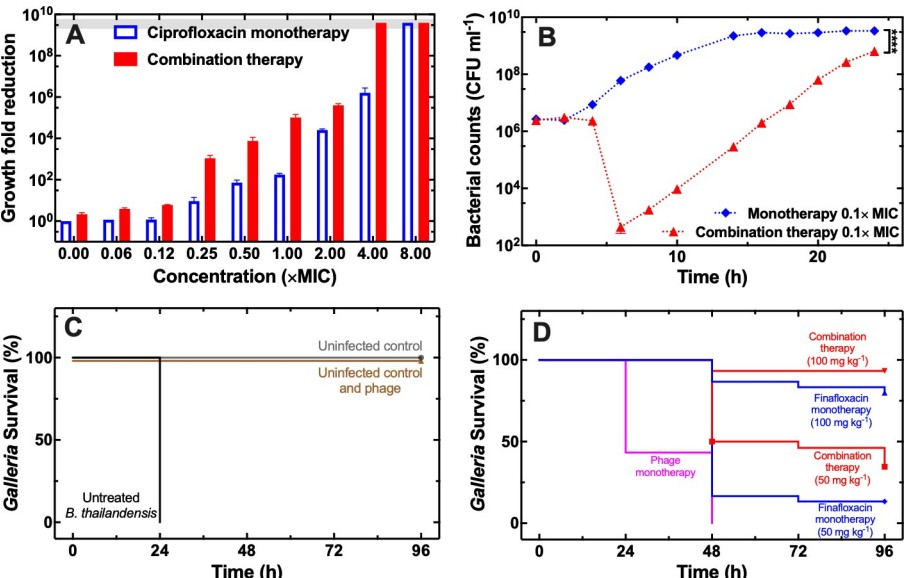

**Fig 5. Synergistic bactericidal effect of phage and ciprofloxacin combination therapy.** (A) Dependence of the ratio between the number of bacteria viable after 24 h incubation in LB medium and the number of bacteria viable 24 h ciprofloxacin monotherapy (blue bars) or phage-ciprofloxacin combination therapy (red bars) on the concentration of ciprofloxacin employed. In all cases the starting inoculum was stationary phase B. thailandensis at a concentration of $5 \times 10^5$ CFU mL$^{-1}$ and bacteria were counted at 24 h via CFU assays. The grey horizontal band represents a survival fold reduction that corresponds to the complete eradication of the bacterial population, i.e. colony counts below limit of detection of 10 CFU mL$^{-1}$. Pair-wise t-tests revealed that combination therapy yielded a significantly (p-value < 0.0001) higher growth reduction at all ciprofloxacin concentrations tested compared to ciprofloxacin monotherapy, apart from 8× MIC. (B) Temporal dependence of bacterial counts following 0.125× MIC ciprofloxacin monotherapy (blue diamonds) or combination therapy with 0.125× MIC ciprofloxacin and phage at an MOI of 1 (red triangles). Symbols and error bars are means and standard errors of the mean of biological triplicates each containing technical triplicates. Very small error bars cannot be visualised due to overlap with the datapoints. Dotted lines are guides-for-the-eye. Survival probability of Galleria mellonella larvae injected with (C) PBS only (uninfected control, grey circles and line), with PBS and $10^5$ phage ml$^{-1}$ (uninfected control and phage, brown triangles and line), with $10^4$ B. thailandensis cells ml$^{-1}$ (untreated B. thailandensis), or with (D) $10^4$ B. thailandensis cells ml$^{-1}$ and either $10^5$ phage ml$^{-1}$ (phage monotherapy, magenta circles and line), or 50 or 100 mg kg$^{-1}$ of finafloxacin (combination monotherapies, red downward triangles or squares and lines) or 50 or 100 mg kg$^{-1}$ of finafloxacin and $10^5$ phage ml$^{-1}$ (blue upward triangles or squares and lines). Each treatment was carried out in biological triplicate, using 10 larvae in each replicate. Numerical values are reported in S7 File.

bactericidal concentration that provided complete eradication of stationary phase *B. thailandensis* (horizontal band in Fig 5A) was 8× MIC for ciprofloxacin monotherapy and 4× MIC for phage-ciprofloxacin combination therapy.

Next, we set out to measure and contrast the dynamics of the bactericidal effect of mono- and combination therapy at a sub-MIC antibiotic concentration. We treated stationary phase *B. thailandensis* with either ciprofloxacin at 0.125× its MIC or a combination of phage at an MOI of 1 and ciprofloxacin at 0.125× its MIC and measured the number of survivors at regular time points via CFU assays (Fig 5B and Data B in S7 File). Following ciprofloxacin monotherapy, the bacterial population was not affected by antibiotic exposure but started expanding after 2h of treatment and reached stationary phase after 14h of treatment (blue diamonds in Fig 5B). In contrast, following combination therapy the bacterial population reduced between 4 h and 6 h and reached a minimum that was >5-log lower compared to the initial bacterial inoculum (red triangles in Fig 5B). The surviving bacterial population started to increase after 6 h of treatment due to the emergence of phage resistance (S6 Fig, Data C and Data D in S7 File) and reached a maximum level at 24 h that was 1-log lower than bacteria treated with ciprofloxacin monotherapy (red triangles and blue diamonds in Fig 5B, respectively). The 5-log reduction in bacterial population after 6h of combination therapy compared to antibiotic monotherapy may provide significant release for the immune system to be able to clear bacterial infections before the emergence of a phage-resistant subpopulation, although further research will be needed to test this hypothesis.

Finally, we set out to measure and contrast the bactericidal effect of mono- and combination therapy in *Galleria mellonella* larvae, a simple infection model which has been increasingly used for microbiological research [44]. We found that ΦBp-AMP1 did not have any toxic effect on the larvae that survived exposure to phage alone as much as uninfected control larvae manipulated with PBS only; in contrast larvae treated with $10^4$ *B. thailandensis* cells ml$^{-1}$ rapidly died (Fig 5C and Data E in S7 File). Phage monotherapy was able to significantly delay death (p-value < 0.0001) and crucially combination therapy with phage and finafloxacin allowed for significantly improved survival compared to finafloxacin monotherapy (p-value = 0.02 for combination vs finafloxacin monotherapy at 50 mg kg$^{-1}$, Fig 5D and Data E in S7 File).

Taken together these data confirm that the use of ΦBp-AMP1 increases antibiotic efficacy against stationary phase *B. thailandensis* both *in vitro* and in a simple *in vivo* infection model.

## Phage-induced downregulation of efflux boosts antibiotic accumulation in *B. thailandensis*

Next, we set out to discover the molecular mechanisms at the basis of the newly found synergistic bactericidal interaction between ΦBp-AMP1 and ciprofloxacin. We treated stationary phase *B. thailandensis* for 4 h either with 0.125× MIC ciprofloxacin monotherapy or phage monotherapy at an MOI of 1 or combination therapy with both 0.125× MIC ciprofloxacin and phage at an MOI of 1. We chose 4 h-long treatments because these treatments returned similar survivor numbers before the > 5-log reduction in survivor numbers measured for combination therapy at the 6 h time point (Fig 5B). We extracted bacterial and phage RNA from biological triplicates of each condition and performed a global comparative transcriptomic analysis [45] among the transcriptomes obtained from these three different conditions and with respect to stationary phase *B. thailandensis* incubated in LB medium for 4 h (S8, S9 and S10 Files). Using principal component analysis, we found that bacterial transcriptome replicates from each condition clustered together and were well separated from replicates from different conditions (S7A Fig and S11 File). The only exception were the bacterial transcriptomes harvested from cultures treated with phage mono- and combination therapy (black circles and red

triangles in S7A Fig) that largely overlapped, suggesting that, at the concentrations employed, phage had a greater impact than ciprofloxacin on the bacterial transcriptomes. Moreover, the phage transcriptomes harvested from cultures treated with mono- and combination therapies also largely overlapped according to our principal component analysis (S7B Fig and S11 File).

Gene ontology enrichment analysis [46] of differentially expressed genes revealed that all treatments investigated resulted in the downregulation of locomotion processes, in accordance with a recent study using *P. aeruginosa* [47], as well as cell localization, cell projection, organelle organization and response to external stimuli processes (Fig 6A, 6B, S12, S13 and S14 Files). Ciprofloxacin monotherapy also caused the downregulation of biological processes involved in cell communication, as well as the upregulation of RNA, catabolic, metabolic and ribosomal processes (Fig 6A and S12 File).

In contrast, both phage monotherapy and combination therapy caused the downregulation of metabolic, membrane transport, translational, energy generation and respiration processes (Fig 6B, S13 and S14 Files). Specifically, genes encoding the major efflux pump BpeEF-OprC

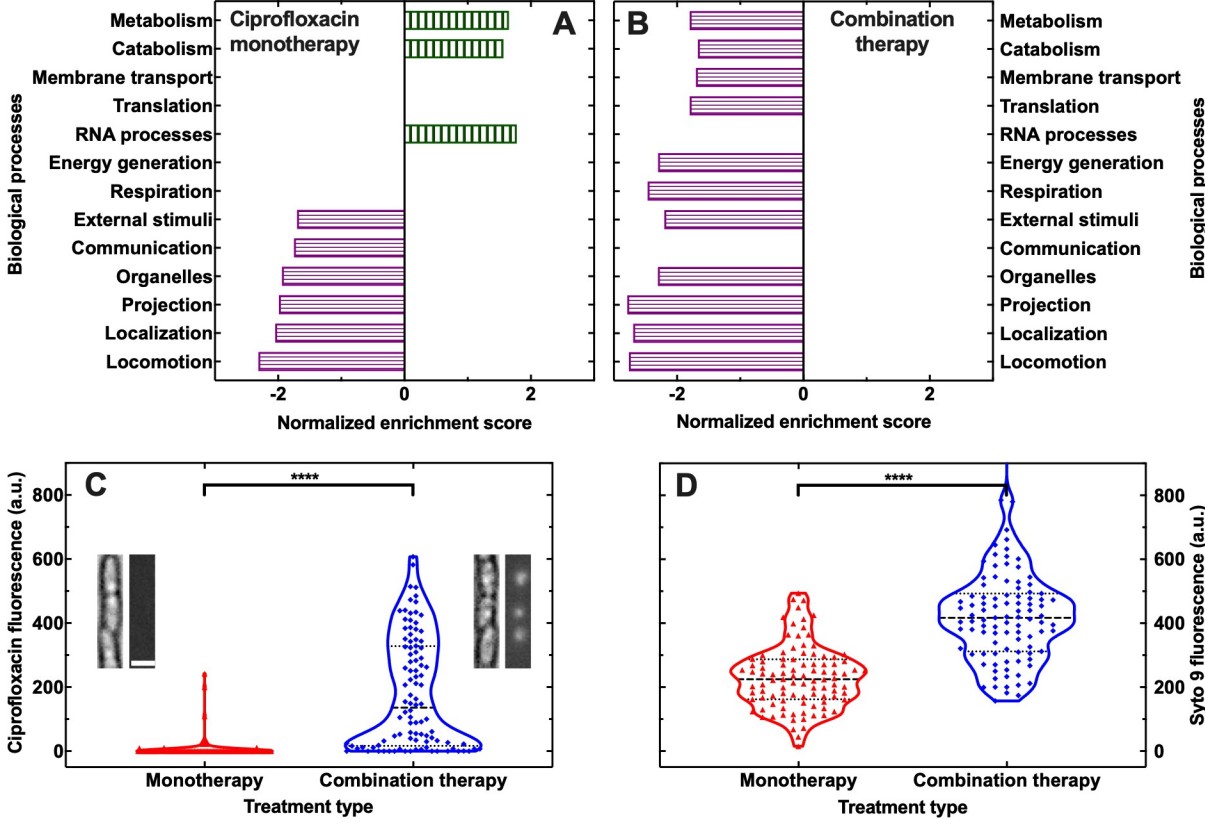

**Fig 6. Synergistic bactericidal interactions between phage and ciprofloxacin are facilitated by phage-induced downregulation of membrane transport in B. thailandensis.** Major biological processes that are significantly enriched and either up- or downregulated (vertically and horizontally patterned bars, respectively) in stationary phase B. thailandensis after 4 h of (A) monotherapy with ciprofloxacin at 0.125× MIC or (B) combination therapy with ciprofloxacin at 0.125× MIC and phage at an MOI of 1 with respect to untreated B. thailandensis incubated for 4 h in LB medium only. Corresponding differential gene expression analyses and gene ontology enrichment analyses are reported in S1–S3 and S4–S6 Files, respectively. If a process is not enriched there is no corresponding data bar. Distribution of fluorescence values of (C) ciprofloxacin-NBD or (D) Syto 9 accumulating in N = 100 individual stationary phase B. thailandensis that had been incubated in either ciprofloxacin-NBD or Syto 9 only or ciprofloxacin-NBD or Syto 9 and phage (red triangles and blue diamonds, respectively). Ciprofloxacin-NBD, Syto 9 and phage were introduced in the microfluidic device at t = 0 at a concentration of 32 μg ml$^{-1}$, 3.34 μM and 10$^{8}$ PFU ml$^{-1}$, respectively. Dashed lines indicate the median of each distribution, dotted lines indicate the quartiles of each distribution. **** indicate a p-value < 0.0001. Insets in (C): representative brightfield and fluorescence images of B. thailandensis after 4h of monotherapy (left) or combination therapy (right). Scale bar: 2 μm. Numerical values are reported in S16 File.

were significantly downregulated along with functional subunits of the $F_0F_1$-type ATP synthases, of the cytochrome O oxidase, of the NADH-quinone oxidoreductase complex, of lipopolysaccharide transport and of secretion systems (S8 Fig and S9 File).

Combination therapy drove a global differential gene regulation that was very similar to that caused by phage monotherapy and gene ontology enrichment analysis did not return any enriched functional category for this comparison. Moreover, the expression of phage encoded genes was also unaffected by the presence of ciprofloxacin (S15 File) and gene ontology enrichment analysis did not return any enriched functional category for this comparison either.

Next, we set out to test the hypothesis that phage-induced downregulation of efflux and lipopolysaccharide transport processes led to increased intracellular accumulation of ciprofloxacin during phage-ciprofloxacin combination therapy. We used our recently reported microfluidics-based time-lapse microscopy platform [48,49] and a fluorescent derivative of ciprofloxacin (i.e. ciprofloxacin-nitrobenzoxadiazole, henceforth ciprofloxacin-NBD) to measure intracellular accumulation of ciprofloxacin in individual stationary phase *B. thailandensis* cells. Using the broth microdilution approach, we determined that the MIC of ciprofloxacin-NBD against stationary phase *B. thailandensis* E264 is 64 μg ml$^{-1}$. Therefore, we chose to deploy ciprofloxacin-NBD at 32 μg ml$^{-1}$, i.e. 0.5× MIC which is within the additive concentration range reported in Fig 3AB. In accordance with our hypothesis, we found that the distribution of ciprofloxacin-NBD fluorescence values after 4 h phage-ciprofloxacin-NBD combination therapy was significantly higher than the distribution of ciprofloxacin-NBD fluorescence values after 4 h ciprofloxacin-NBD monotherapy (Fig 6C and S16 File). Similarly, we found that the distribution of Syto 9 (i.e. a substrate of efflux pumps [50]) fluorescence values after 1 h of supply in the presence of phage was significantly higher than the distribution of Syto 9 fluorescence values after 1 h of supply in the absence of phage (p-value < 0.0001, Fig 6D and S16 File).

Moreover, using this platform we did not find evidence of either cell filamentation or a significant difference in cell size between stationary phase *B. thailandensis* incubated in LB growth medium only (S9A–S9E Fig and S17 File), or during phage monotherapy (S9F–S9J Fig) or during combination therapy with phage at an MOI of 1 and ciprofloxacin at 0.125× MIC (S9K–S9O Fig). We found that stationary phase *B. thailandensis* infected with phage in the presence of sub-inhibitory concentrations of ampicillin or ciprofloxacin produced less phage particles than in the absence of antibiotics in planktonic cultures and plaques of similar size on agar cultures (S1 and S10 Figs, respectively). Sub-inhibitory concentrations of trimethoprim also led to smaller plaques with phage propagation starting significantly later in the presence of trimethoprim (S1 and S10 Figs and S18 File). We also did not observe plaque formation on LB agar plates in the presence of sub-inhibitory concentrations of ciprofloxacin in the absence of phage ΦBp-AMP1. Moreover, in our phage transcriptome analysis we found evidence of expression of structural phage proteins, but we did not find evidence of expression of genes encoding excisionases, i.e. proteins required for the excision of dormant phage from within the hosts genome [51], neither during phage mono- nor combination therapy (S15 File). However, it is conceivable that one or more of the 11 hypothetical proteins expressed by ΦBp-AMP1 (S15 File) could perform excisionase activity.

Taken together these data demonstrate that the observed phage-antibiotic interactions are not due to cell filamentation, increased phage particle production or phage induction in the presence of antibiotics but are instead facilitated by phage-induced downregulation of membrane transport and energetic processes that are involved in the permeability to and efflux of ciprofloxacin leading to higher intracellular ciprofloxacin accumulation in the presence of phage.

## Discussion

### Emergence of resistance to phage

Phage and bacteria are engaged in a constant arms race leading to the evolution of a multitude of non-mutually exclusive antiphage defence mechanisms, including the well-understood phage receptor alteration, restriction-modification, abortive infection, CRISPR-Cas systems, as well as newly discovered defence systems [52–54]. It is well established that increased phage virulence selects for the evolution of host resistance if the costs associated with resistance are outweighed by the benefits of the capability to avoid infection [55,56]. Accordingly, we found that the level of resistance to phage increased with the strength of phage predation. Interestingly, even within a putatively clonal *B. thailandensis* population we found evidence of emergence of different levels of resistance due to mutations in eight different genes. Three of these genes encoded glycosyltransferase and O-antigen synthesis, that are strongly linked with the LPS [57], and capsular polysaccharides. Our data therefore suggest that the LPS or capsular polysaccharides could be the receptors for phage ΦBp-AMP1. This hypothesis is further corroborated by our global comparative transcriptomics analysis demonstrating the downregulation of LPS assembly associated genes following exposure to ΦBp-AMP1. Indeed, the LPS is a well-known receptor for many bacteriophages and mutations of LPS confer resistance to phage in a variety of bacteria [58–61], whereas capsular polysaccharides have recently been shown to serve as primary receptors of *E. coli* phage [62,63].

### Phage-antibiotic interactions

The interactions between two or more antimicrobials as components of combination therapies are broadly classified in three main types: additive (the sum of the effect of each component), synergistic (a larger-than-additive effect) and antagonistic (a smaller-than-additive effect [15,64,65]). In the context of phage-antibiotic therapy instead, phage-antibiotic synergy has been defined as the stimulation of phage replication when bacteria are treated with sub-inhibitory concentrations of antibiotics [32,33,66]. Considering that in accordance with a recent report [67] we did not find stimulation of phage replication in the presence of antibiotics, we chose to use the more broadly accepted definitions of additive, synergistic and antagonistic interactions, introduced above [15,64,65]. Specifically, by using statistical analysis on interaction plots we found an additive effect between the phage and most of the tested molecules from the quinolone, β-lactam and tetracycline antibiotic classes, antagonism between the phage and tetrahydrofolate synthesis inhibitors and bactericidal synergy between the phage and the quinolone ciprofloxacin. These effects were comparable or more efficient in suppressing bacterial growth than previously reported synergistic phage-antibiotic effects [32,38,39]. For example, ceftazidime efficacy increased by a factor of 2 compared to ceftazidime monotherapy when used in combination with the podovirus vB_BpP_HN01 [68] or the myovirus KS12 [1] against *B. pseudomallei* or *B. cenocepacia*, respectively; we measured an 8-fold increase in ceftazidime efficacy against *B. thailandensis* in the presence of phage ΦBp-AMP1.

Using the temperate phage HK97 in combination with ciprofloxacin, a previous study has reported bactericidal synergy against *E. coli* K12, with complete eradication being achieved at 0.5× MIC ciprofloxacin [36]; we measured a comparatively weaker bactericidal synergy between ΦBp-AMP1 and ciprofloxacin, with complete eradication being achieved at 4× MIC ciprofloxacin. However, it is worth noting that stationary phase *B. thailandensis* is significantly more resistant than exponential phase *E. coli* K12 with ciprofloxacin MIC values of 2 and 0.2 μg mL$^{-1}$, respectively.

Since ΦBp-AMP1 was previously described to effectively lyse different strains of both *B. thailandensis* and *B. pseudomallei* [40–42], we expect that combination treatments using ΦBp-AMP1 will boost antibiotic efficacy also against *B. pseudomallei*, i.e. the causative agent of meliodosis, although we recognise that further research is needed to optimise treatments against this closely related species.

## Factors affecting phage-antibiotic interactions

The dependence of phage-antibiotic interactions on the antibiotic class is well established [32,33,39,66,67,69], however, the mechanisms at the basis of this dependence remain largely unknown. For example, the myovirus KS12 broadly synergised with quinolones, β-lactams and tetracycline when used to inhibit growth of *B. cenocepacia* but antagonised with aminoglycosides [33]. The myovirus ΦHP3 displayed a synergistic effect with ceftazidime, an additive effect with kanamycin and an antagonistic effect with chloramphenicol when used to inhibit growth of *E. coli* [39]. The phage PYO[SA] antagonised tetracycline, azithromycin, and linezolid but synergised with daptomycin, vancomycin and kanamycin when used to inhibit growth of *S. aureus* [38]. Here we advance this understanding by showing that even molecules within the same class can display a dramatically different extent and range of additive interactions with the same phage and that when used with phage the same molecule can simultaneously display an additive effect in inhibiting bacterial growth and a synergistic effect in killing bacteria.

Moreover, the interactions between antibiotics and phage are often studied with only one or two concentrations of the antimicrobials, which are insufficient in predicting combinatorial concentrations that are effective during treatment [24]. By assessing bacterial growth when exposed to multiple orders of magnitude of antibiotic concentrations and phage titers, we discovered that phage-antibiotic interactions strongly depend on the antibiotic concentration employed, but surprisingly do not vary with phage titer, a key difference from previous findings [39]. These newly discovered dependences should be taken into account when designing rational phage-antibiotic therapy [24,25]. In fact, our data suggest that it might be relatively straightforward to hit a suitable ΦBp-AMP1 phage titer in an *in vivo* setting, where phage are broadly tolerated [70] but their pharmacokinetic and pharmacodynamic parameters are less known compared to antibiotics [71].

Finally, a further factor often discussed to leverage phage-antibiotic synergy is the impact of sequential deployment of these antimicrobials. This approach has shown prior success, for example, against multidrug resistant *Pseudomonas aeruginosa* [37]. Considering that *B. thailandensis* downregulates efflux during exposure to ΦBp-AMP1, we anticipate that phage administration followed by antibiotic administration could produce a stronger bactericidal effect than simultaneous administration. However, we recognise that further work will be required to investigate the potential benefit of staged administration.

## Mechanistic understanding of phage-antibiotic interactions

The additive effects between phage and antibiotics are stronger and broader in our experiments compared to previous reports [33,66,68], possibly due to differences at the phage level, i.e., a podovirus vs a myovirus, at the bacterial strain and physiology level, i.e., stationary phase *B. thailandensis* vs exponential phase *B. cenocepacia*, or due to a different mechanism of interaction between phage and antibiotics. Indeed, KS12 and a variety of other phage displayed an increase in plaque size and phage titer in the presence of sub-inhibitory concentrations of antibiotics [30,32,35,66], possibly caused by the acceleration of phage assembly and cell lysis due to cell filamentation in the presence of antibiotics [32,33,39,72]. In contrast, we did not find evidence neither of cell filamentation nor of an increase in plaque size and phage titer in the

presence of sub-inhibitory concentrations of quinolones, β-lactams or tetracyclines. Reduced phage titer in the presence of sub-inhibitory concentrations of trimethoprim could instead explain the observed antagonism between ΦBp-AMP1 and trimethoprim. Moreover, temperate phage activity is known to enhance antibiotic efficacy through depletion of lysogens [36]. Although we found evidence of prophage ΦE125 and ΦE12-2 [2,3], it is unlikely that depletion of ΦE125 or ΦE12-2 lysogens plays a role in our experiments since we did not find evidence of plaque formation in the presence of ciprofloxacin, that is a lysogen activating antibiotic [36], in the absence of ΦBp-AMP1.

Based on our global gene expression analysis, we hypothesized that antibiotics are more effective in inhibiting *B. thailandensis* growth in the presence of ΦBp-AMP1 due to increased antibiotic efflux out of the cell. Indeed, the multi-drug efflux pump BpeEF-OprC [10] was downregulated in the presence of phage and it is known that deletion of BpeEF-OprC causes increased susceptibility to quinolones and β-lactams [11]. Moreover, we detected a strong downregulation of genes associated with aerobic respiration and transmembrane transport of protons, effectively reducing the availability of ATP for active transport of substrates as well as the proton motive force. A reduction in proton motive force levels leads to reduced antibiotic efflux [73–75], and in the long-term selection for mutations in drug efflux components [76]. In accordance with our hypothesis, we found that a fluorescent derivative of ciprofloxacin accumulates in individual *B. thailandensis* cells at significantly higher levels in the presence of ΦBp-AMP1 compared with in its absence. Noteworthy, previous studies using phage that have the efflux component TolC as a receptor, demonstrated that emergence of phage resistance in bacteria via *tolC* mutations led to increased antibiotic susceptibility [60,77]. However, it is also conceivable that decreased lipopolysaccharide transport in the presence of phage leads to increased membrane permeability to ciprofloxacin and thus further contributes to enhanced antibiotic accumulation in the presence of phage. Finally, it is also conceivable that the phage-antibiotic interactions we observed are also due to phage and antibiotics targeting cells in different metabolic states as recently hypothesised [67].

## Conclusions

The interactions between phage and bacteria are multifaceted and complex due to billion-year long arms race between both entities [78–82] and are further modified when a second selective pressure is imposed, such as the presence of antibiotic compounds secreted by other microbes in the environment [83]. Understanding such interactions might hold the key for successful antimicrobial therapy and to overcome the current antimicrobial resistance crisis. Considering that stationary phase bacteria are traditionally refractory to antibiotics, especially in spatial structures such as biofilms where antibiotic diffusion is further hindered [48], our data offer a potential route for their eradication by combining low doses of clinically relevant antibiotics with low doses of phage, that should be easily obtainable *in vivo* thanks to the self-propagating nature of phage.

## Methods

### Bacterial and bacteriophage strains

*B. thailandensis* E264 strain and the temperature dependent, lytic bacteriophage ΦBp-AMP1 were obtained from the Department of Immunology, Faculty of Medicine Siriraj Hospital, Mahidol University.

## Bacterial culturing

*B. thailandensis* was stored at -80˚C and streaked on Lysogeny broth (LB, 10 g/L Tryptone, 5 g/L Yeast extract, 10 g/L NaCl, Melford) agar plates (10 g/L, 1.5% Agar) every two weeks. Overnight cultures were setup by inoculating a single *B. thailandensis* colony from a plate into flasks containing 50 mL of LB broth and were grown for 17 h at 37˚C on shaking platforms set at 200 rpm.

## Propagation and titration of phage

Phage propagation was carried out as previously reported [84]. Briefly, overnight cultures of *B. thailandensis* E264 were diluted 1000× in 50 mL LB medium to obtain a bacterial concentration of approximately $2\times10^6$ CFU mL$^{-1}$. These sub-cultures were then incubated for 4 h at 37˚C and 200 rpm, allowing them to reach early exponential phase (~$8\times10^6$ CFU mL$^{-1}$). Then, these sub-cultures were infected with ΦBp-AMP1 at an MOI = 0.01 and incubated for 17 h at 37˚C and 200 rpm. On the following day, cells were pelleted by centrifugation for 40 minutes at 3000 *g* and the phage-containing supernatant was filtered twice (Sartorius Minisart 0.2 μm) to obtain a phage stock. The phage concentration within this stock was then determined via the double agar overlay technique [83]. Briefly, LB top agar (10 g/L, 0.5% Agar) was melted in a microwave and allowed to cool to below 40˚C. An aliquot of an overnight *B. thailandensis* E264 culture was added to the melted top agar at a volume: volume concentration of 1:50. LB agar plates were subsequently coated in a thin layer of the top agar above creating a continuous bacterial lawn. In parallel, a 10-fold dilution series of the phage stock was prepared in LB medium. Phage was then titred using the standard double agar overlay technique [4]. All plates were then incubated at 37˚C for 17 h after which the phage induced plaques in the bacterial lawn were counted and the plaque forming units (PFU) per mL$^{-1}$ of the phage stock was calculated. Phage stocks were stored at 4˚C for a maximum of two weeks before a new propagation was performed. Prior to each use, the phage concentration within the phage stock was determined via the double agar overlay technique above.

## Bacterial growth in the presence of phage

In order to measure the growth of bacteria in the presence of phage, we used two independent approaches. Firstly, we measured the concentration of bacteria during phage infection over time using colony forming unit (CFU) assays [85]. Briefly, three overnight cultures of *B. thailandensis* were diluted in 50 mL LB medium to obtain a bacterial concentration of approximately $2\times10^6$ CFU mL$^{-1}$. ΦBp-AMP1 was added to these three sub-cultures at an MOI of 1 and the sub-cultures were incubated at 37˚C and 200 rpm. Triplicate aliquots were taken from each infected sub-culture every two hours, serially diluted in LB and plated on LB agar plates. These plates were incubated 37˚C for 24h before counting CFU on each plate to calculate the bacterial concentration at each time point in each sub-culture as the mean and standard error of the mean of biological triplicate each containing technical triplicate. The growth of control uninfected cultures was measured in a similar manner without the addition of phage. Secondly, we measured the change of bacterial density over time during phage infection by measuring the optical density of growing sub-cultures. Briefly, three overnight cultures of *B. thailandensis* were diluted in LB in the wells of a 96 well plate and mixed with phage to obtain a final bacterial concentration of $5\times10^5$ CFU mL$^{-1}$ and a final phage MOI of either $10^{-4}$, $10^{-2}$, 1 or $10^2$. The plates were the incubated at 37˚C and 200 rpm and the optical density of each well was measured every 10 min via a CLARIOstar plate reader system (BMG) and blank corrected to the optical density measured in wells containing LB medium only. Each condition was tested in 84 technical replicates obtained from biological triplicate. This same approach

was employed to measure the re-growth of low- and high-resistant mutants harvested from the experiments above in the presence of phage at an MOI of 1.

## Determination of heritable resistance to phage

In order to determine whether heritable resistance to phage had emerged in bacteria from wells where we measured bacterial growth in the presence of phage, we re-inoculated these survivors in wells of a 96-well plate containing LB and phage at a concentration of ~$5\times10^5$ PFU mL$^{-1}$. We incubated these plates at 37°C and 200 rpm for 24 h and measured the optical density as described above. We employed this same approach to determine whether heritable resistance to phage had emerged in bacteria from wells where we measured bacterial growth in the presence of both phage and each of the antibiotics reported in Table 1.

## Determination of antibiotic minimum inhibitory concentrations and the impact of phage on antibiotic efficacy

All antibiotics employed in this study were purchased from Merck. The minimum inhibitory concentration (MIC) of each antibiotic employed was determined via the broth dilution method [86]. Briefly, each antibiotic was dissolved according to the manufacture's specifications and diluted in a two-fold dilution series in LB medium within wells of a 96-well plate. Stationary *B. thailandensis* E264 bacteria were then added at a final concentration of $5\times10^5$ CFU mL$^{-1}$ to each well. The plates were incubated at 37°C and 200 rpm and after 24 h the optical density of each well was measured via a CLARIOstar plate reader system (BMG) and blank corrected to the optical density measured in wells containing LB medium only. The minimum inhibitory concentration was determined as the minimum concentration of antibiotic for which we measured an optical density value that was less than 10% of the optical density value that we measured in wells containing LB medium and bacteria only. All MIC assays were performed at least in biological and technical triplicate from which mean and standard error of the mean were calculated. Next, to determine the influence of ΦBp-AMP1 on the efficacy of each antibiotic, the experiments above were repeated with the addition of ΦBp-AMP1 at an MOI of 1 (i.e. $5\times10^5$ PFU mL$^{-1}$).

## Phage growth in the presence of antibiotics

Overnight cultures of *B. thailandensis* E264 were diluted 1000× in 50 mL LB medium to obtain a bacterial concentration of approximately $2\times10^6$ CFU mL$^{-1}$. ΦBp-AMP1 was added to these sub-cultures at an MOI of 1 and the sub-cultures were incubated at 37°C and 200 rpm. In separate experiments, ΦBp-AMP1 was added at an MOI of 1 together with either ampicillin or ciprofloxacin or trimethoprim at 0.25× their respective MIC. 1 mL triplicate aliquots were taken from the infected sub-cultures at two-hour intervals for twenty-four hours and the phage propagation over time was monitored by determining plaque forming units per millilitre via the double agar overlay technique above. Each experiment was then performed in biological triplicate.

## Determination of time-dependent bacterial growth in the presence of antibiotic and phage

Overnight cultures of *B. thailandensis* E264 were diluted to a concentration of ~$5\times10^5$ CFU mL$^{-1}$ in LB medium in wells of a 96-well plate, together with either ΦBp-AMP1 at an MOI of 1, or levofloxacin, cefaclor or trimethoprim at 0.25× their respective MIC, or both ΦBp-AMP1 at an MOI of 1 and either levofloxacin, cefaclor or trimethoprim at 0.25× their respective MIC.

Each plate was then incubated for 24 h in a CLARIOstar plate reader (BMG) at 37˚C and 200 rpm, measuring optical density at $\lambda$ = 600nm ($OD_{600}$) every 30 min. Each experiment was repeated in biological and technical triplicate from which we calculated mean and standard error of the mean of each measurement.

## Bacterial killing assays

Bacterial killing assays were performed as previously reported [45]. Briefly, overnight cultures of *B. thailandensis* E264 were diluted 1000× in 50 mL fresh LB growth medium and infected with either solely ciprofloxacin (at a concentration between 0.625× and 8× the MIC), or a combination of ciprofloxacin and phage at MOI = 1. The cultures were incubated at 37˚C and 200 rpm for 24 h after which the colony count (CFU mL$^{-1}$) was determined as reported above. We quantified bactericidal efficacy of each treatment as the ratio of the colony counts in untreated control experiments over the colony counts measured in each treatment.

## Determination of heritable resistance to antibiotics

In order to determine whether heritable resistance to antibiotics had emerged in bacteria from wells where we measured bacterial growth in the presence of phage and antibiotics, we re-inoculated these survivors in wells of a 96-well plate containing LB and either phage at a concentration of ~5×10$^5$ PFU mL$^{-1}$, or the antibiotic employed at a concentration in range 0.125–128 µg ml$^{-1}$, or both phage at a concentration of ~5×10$^5$ PFU mL$^{-1}$ and the antibiotic employed at a concentration in range 0.125–128 µg ml$^{-1}$. We incubated these plates at 37˚C and 200 rpm for 24 h and measured the optical density as described above.

## Determination of the impact of antibiotics on plaque size

To determine whether phage plaque sizes may be influenced by the presence of antibiotics, we implemented a previously reported protocol [32]. Briefly, LB top agar was melted in a microwave and allowed to cool below 40˚C. An aliquot of an overnight *B. thailandensis* E264 culture was added to the melted top agar at a volume: volume concentration of 1:50 alongside phage at a concentration of 10$^1$ PFU mL$^{-1}$ and either ciprofloxacin, trimethoprim or ampicillin at 0.5× and 0.125× of their respective monotherapy MICs. The resulting mixture was added to the top agar. The top agar was then well mixed by inversion, plated evenly onto LB agar plates (10 g/L, 1.5% Agar) and incubated for 24 h at 37˚C. The plates were then imaged using a Xiaomi Mi A2 mobile phone camera. The plaque size was determined via image analysis of 120 plaques for each condition in biological triplicate using the ImageJ software.

## Evaluation of mono- and combination therapy in the *Galleria mellonella* infection model

Larvae weighing 250–350 mg were collected the day prior experiment in petri dishes and kept with no food in the dark at 30˚C. The next day, they were cleaned and removed from the silk by quick washing with 70% ethanol. 1 mL of E264 *B. thailandensis* overnight culture was centrifuged at 13,000 rpm for 15 mins. The supernatant was removed, and the pellet was washed with PBS. This process was repeated, and the $OD_{600}$ was measured and adjusted to reach a bacterial density of 10$^4$ bacteria mL$^{-1}$. Using a 22-gauge Hamilton syringe with a bevel tip, in each treatment 10 larvae were injected into the bottom right proleg with either PBS or 10$^4$ *B. thailandensis* cells mL$^{-1}$. This was followed by a second injection 2 h post-infection (P.I.) into the bottom left proleg with either PBS, 10$^5$ phage ml$^{-1}$, 10$^4$ *B. thailandensis* cells mL$^{-1}$, 10$^4$ *B. thailandensis* cells mL$^{-1}$ and 10$^5$ phage ml$^{-1}$, 10$^4$ *B. thailandensis* cells mL$^{-1}$ and 50 or 100 mg kg$^{-1}$

of finafloxacin, $10^4$ *B. thailandensis* cells mL$^{-1}$ and 50 or 100 mg kg$^{-1}$ of finafloxacin and $10^5$ phage ml$^{-1}$. The needle was washed with 70% ethanol, followed by PBS, when switching to a new treatment group to avoid cross-contamination. The larvae were then placed in petri dishes and incubated at 37°C in the dark. The number of dead larvae was recorded every 24 h over a 4-day period. Larvae were considered dead when they showed no response to touch. All experiments were repeated in biological triplicate, with the data pooled to give n = 30 larvae. Survival data were plotted as a Kaplan-Meier graph, with statistical significance between groups tested using the log-rank (Mantel-Cox) test.

## Determination of single-cell morphology and ciprofloxacin accumulation

To measure the size and morphology of individual bacterial during phage-monotherapy or antibiotic-phage combination therapy, we deployed the microfluidic mother machine device as previously described [87]. Briefly, the device consists of a central channel that measures 25 μm and 100 μm in height and width, respectively, and six thousand lateral side channels, each 1 μm in width and height and 25 μm in length [88] and was fabricated in polydimethylsiloxane (PDMS, Sylgard 184 silicone elastomer kit, Dow Corning) following our previously reported protocols [89]. Bacteria were prepared by pelleting an overnight culture of *B. thailandensis* for 15 minutes at 3000 *g*. The resulting bacterial pellet was resuspended at a previously optimised nominal OD$_{600}$ of 50 in medium obtained by double filtering the supernatant from the spun down culture using 0.22 μm filters [90,91]. These bacteria were then introduced in the central channel of the mother machine device from where they reached the lateral channel at an average concentration of one bacterium per channel [92]. Next, the device was mounted on an inverted microscope (IX73 Olympus, Tokyo, Japan) located in a temperature-controlled chamber kept at 37°C [93]. Fluorinated ethylene propylene tubing (1/32" × 0.008") was connected to the device as inlet and outlet tubes further connected to a computerised pressure-based flow control system (MFCS-4C, Fluigent) [94]. Next, LB medium only or LB medium containing 2×10$^8$ PFU mL$^{-1}$ phage or LB medium containing 2×10$^8$ PFU mL$^{-1}$ phage and 0.125× MIC ciprofloxacin was continuously supplied in the device at a constant flow rate of 100 μl/h. Simultaneously, bright field images of 20 areas of the mother machine, each containing 23 lateral channels were acquired at 2 min intervals via a 60× 1.2 N.A. objective (UPLSA-PO60XW, Olympus) and an sCMOS camera with an exposure time of 0.01 s (Zyla 4.2, Andor, Belfast, United Kingdom) controlled via Labview [95]. This microfluidics-microscopy platform was also used to quantify the accumulation of ciprofloxacin or Syto 9 as previously reported [96,97]. Briefly, LB medium containing a fluorescent ciprofloxacin derivative, i.e. ciprofloxacin-nitrobenzoxadiazole (ciprofloxacin-NBD) or ciprofloxacin-NDB and 2×10$^8$ PFU mL$^{-1}$ phage was continuously supplied in the device at a constant flow rate of 100 μl/h. In both cases ciprofloxacin-NBD was supplied at a concentration of 32 μg ml$^{-1}$. Similarly, LB medium containing Syto 9 or Syto 9 and 2×10$^8$ PFU mL$^{-1}$ phage was continuously supplied in the device at a constant flow rate of 100 μl h$^{-1}$. In both cases Syto 9 was supplied at a concentration of 3.34 μM as previously described [98]. Bright field images were acquired as described above together with corresponding fluorescence images acquired by exposing the bacteria for 0.03 s to the blue excitation band of a broad-spectrum LED (CoolLED pE300white, power = 8 mW at the sample plane, Andover, UK) via a FITC filter [99]. All images were analysed using the ImageJ software as previously described [100].

## Determination of mutations that enable resistance to phage

In order to determine the mutations that enable resistance to phage in bacteria from wells where we measured bacterial growth in the presence of phage, we plated these survivors on LB

agar plates, incubated these plates at 37˚C for 24h and shipped the plates to MicrobesNG. MicrobesNG performed DNA isolation and ull genome sequencing using 2×250 bp paired end sequencing with a minimum coverage of 30× on the Illumina HiSeq sequencer. Analysis of the sequenced genomes was performed using the *breseq* pipeline [101].

## Comparative bacterial and phage transcriptomic analysis

RNA isolation, library preparation, sequencing, and transcriptomic data processing was performed as previously reported [102]. Briefly, RNA isolation was performed using the RNeasy Mini kit (QIAGEN), according to the manufacturer's specifications. Bacteria were grown in triplicate for 4h in flasks containing 50 mL LB, or LB containing phage at an MOI of 1, or LB containing ciprofloxacin at 0.125× MIC, or LB containing both phage at an MOI of 1 and ciprofloxacin at 0.125× MIC. DNA removal during extraction was carried out using RNase-Free DNase I (Qiagen). RNA concentration and quality were measured using Qubit 1.0 fluorometer (ThermoFisher Scientific) and 2200 TapeStation (Agilent), respectively, and only samples with an RNA integrity number above 8 were sequenced using Illumina NovaSeq 6000. Transcript abundance was quantified using Salmon for each gene in all samples. Subsequent differential analysis was performed using DEseq2 in R software to quantify the log2 fold change in transcript reads for each gene and compared across the four different experimental conditions. Significantly differentially expressed genes were defined as having a log2 fold change greater than 1 and a p-value adjusted for false discovery rate of $< 0.05$. Gene Set Enrichment Analysis was performed using the clusterProfiler package for R [103]. Enrichments in terms belonging to the "Biological process", "Molecular function" or "Cellular component" ontology were calculated by ranking the genes by differential expression and calculating an Enrichment Score (a weighted Kolmogorov–Smirnov-like statistic) for each ontology term. P values were adjusted for false discovery by using the method of Benjamini and Hochberg [104]. Finally, the lists of significantly enriched terms were simplified to remove redundant terms, as assessed via their semantic similarity to other enriched terms, using clusterProfiler's simplify function [105].

## Statistical non-linear regression model

We developed a statistical non-linear regression model to fit the experimental data in order to estimate treatment output and the associated uncertainty also for treatment conditions that we did not investigate experimentally. Since the distribution of bacterial growth values (in terms of $OD_{600}$) in the presence of phage was skewed, we fitted our data with a Gamma distribution function. Moreover, to account for inaccuracies in optical density measurements via a plate reader, we assumed any measurement below an optical density of 0.05 to be equal to 'zero'. Hence, the data had a non-trivial number of zero values, which could not be captured by a conventional gamma distribution. To accommodate this, we included a parameter which controls the probability of an experiment returning an optical density equal to zero. This two-part distribution strategy resulted in what is often called a "hurdle model" [106]. This distribution then had three parameters, one that represented the probability of measuring a value of zero, and two representing the gamma distribution, in our case parameterised to have a mean parameter and a shape parameter. We assumed that these three parameters could change with the antibiotic concentration and with the phage concentration. We also assumed these relationships to be potentially non-linear. In other words, we assumed the probability of observing a value of zero, the average non-zero observation, and the spread of non-zero observations all depended on the antibiotic and phage concentrations in a complex manner. As such, we modelled these parameters using cubic regression tensor product smoothing splines [107]. Additionally, the spline knot locations for the two gamma distribution parameters were

constrained to only the points where a non-zero observation was made. To fit this model, and estimate the various parameters, including splines, we used Markov Chain Monte Carlo (MCMC), using the brms R package [108], with default options used for the priors. This provided us with posterior distributions for all the unknown parameters, allowing us to make probabilistic statements and provide full uncertainty estimates. To better capture the uneven measurement intervals of empirical data and because we were interested in the relationships between antibiotic and phage concentrations, it was convenient to apply a log2 transform to the antibiotic concentration and a log10 transformation to the phage concentration values. To avoid numerical issues when the antibiotic or phage concentrations were equal to 0, we added a small value to each zero before applying the transformation (0.01 for the antibiotic data, and $1e^{-10}$ for the phage measurement data). The model allowed us to continuously predict bacterial growth in terms of optical density and the associated uncertainty for both treatment concentrations that were experimentally investigated as well as treatment concentrations that were not experimentally investigated.

## Supporting information

**S1 Fig. Phage amplification in the presence of sub-inhibitory concentrations of antibiotics.** Temporal dependence of phage counts from stationary phase B. thailandensis cultures incubated in LB medium only (black circles), or in LB medium containing ampicillin (green triangles), ciprofloxacin (red squares) or trimethoprim (purple diamonds) at 0.25× their respective MIC values. In all cases the initial bacterial inoculum was $2 \times 10^6$ CFU ml$^{-1}$ and the starting concentration of phage was $2 \times 10^6$ PFU ml$^{-1}$. Symbols and error bars are means and standard errors of the mean of phage count measurements from biological and technical triplicates. Very small error bars cannot be visualised due to overlap with the datapoints. Dashed lines are guides-for-the-eye. **** indicate a p-value < 0.0001. Numerical values are reported in S2 File. (TIFF)

**S2 Fig. Dependence of bacterial density on phage MOI.** Distribution of B. thailandensis density values, measured in OD$_{600}$, after 24 h exposure to phage at an MOI of (A) $10^{-4}$, (B) $10^{-2}$, (C) $10^0$ or (D) $10^2$. For each condition, bacterial density measurements were carried out in 84 independent micro-cultures from biological triplicates. Numerical values are reported in Data C in S1 File. (TIFF)

**S3 Fig. B. thailandensis growth in the presence of phage at 25˚C.** Bacterial density measurements after 72 h incubation at 25˚C in the presence of phage at an MOI of 1. Each black circle represents a bacterial density value performed on one of 84 technical micro-culture replicates from biological triplicates. The green vertical band represents the mean and standard error of the mean of corresponding bacterial density measurements in the absence of phage. Numerical values are reported in Data C in S1 File. (TIFF)

**S4 Fig. Regrowth of resistant mutants in the presence of phage.** Growth of (A) high-resistant and (B) low-resistant mutants when re-inoculated in 96-well plates in the presence of phage at an MOI of 1 for 48h. Data points for each mutant have been color coded as in Fig 1C and 1D, HR and LR stands for high-resistant and low-resistant, respectively, and mutants have been numbered in descending order of resistance according to the data in Fig 1C, i.e. HR1 is the highest resistant mutant whereas LR5 is the lowest resistant mutant investigated. Green triangles represent the growth of the parental strain in the absence of phage. Symbols and error bars are means and standard errors of bacterial density values, measured in OD$_{600}$, obtained

from 9 technical replicates from biological triplicates. Corresponding correlation between (C) the final optical density at the end of the second 48h exposure to phage and the optical density measured for each survivor population at the end of the first 48h exposure to phage and (D) the onset of growth during the second exposure to phage and the optical density measured for each survivor population at the end of the first 48h exposure to phage. The red lines are linear regressions to the data. Numerical values are reported in S3 File.
(TIFF)

**S5 Fig. Interactions between ΦBp-AMP1 and folate synthesis inhibitors.** Experimental data (symbols) and model predictions (lines and bands) describing the dependence of bacterial density on the concentration of (A) sulfamethoxazole and (C) trimethoprim/sulfamethoxazole (at a ratio of 1:5) in the absence (blue diamonds) and presence of phage ΦBp-AMP1 at an MOI of 1 (red triangles) after 24 h treatment. Each symbol represents the bacterial density measured in one of 15 technical replicates collated from biological triplicates. Some of the symbols overlap with each other. The lines and shaded areas are the medians, upper and lower quartiles, estimated by fitting our statistical non-linear regression model to our experimental data via Markov Chain Monte Carlo simulations. Corresponding predicted probability of an additive interaction between phage and (B) sulfamethoxazole and (D) trimethoprim/ sulfamethoxazole (ratio 1:5) is below 90%. Not shaded or blue shaded areas indicate antibiotic concentration ranges where the phage or the antibiotic dominate, respectively. Purple shaded areas indicate antagonism. Numerical values are reported in Data B in S5 File.
(TIF)

**S6 Fig. Resistance to phage and ciprofloxacin.** (A) Dependence of bacterial density, measured in $OD_{600}$, on the concentration of ciprofloxacin for B. thailandensis that had not undergone any previous therapy (green squares), B. thailandensis that had undergone a 24 h monotherapy with ciprofloxacin at 0.125× its MIC (blue diamonds), B. thailandensis that had undergone a 24 h monotherapy with phage at an initial MOI of 1 (black circles), or B. thailandensis that had undergone a 24 h combination therapy with ciprofloxacin at 0.125× its MIC and phage at an initial MOI of 1 (red triangles). Symbols and error bars are means and standard errors of the means of bacterial density measurements obtained from biological triplicates each containing five technical micro-culture replicates. Very small error bars cannot be visualised due to overlap with the datapoints. Dashed lines are guides-for-the-eye. The horizontal dashed line represents 10% of the bacterial density value measured after 24 h incubation in LB medium in the absence of ciprofloxacin and phage. (B) Bacterial density, measured in $OD_{600}$, after 24 h exposure to phage at an initial MOI of 1 for B. thailandensis that had not undergone any previous therapy (green squares), B. thailandensis that had undergone a 24 h monotherapy with ciprofloxacin at 0.125× its MIC (blue diamonds), B. thailandensis that had undergone a 24 h monotherapy with phage at an initial MOI of 1 (black circles), or B. thailandensis that had undergone a 24 h combination therapy with ciprofloxacin at 0.125× its MIC and phage at an initial MOI of 1 (red triangles). Each symbol represents a bacterial density measurement obtained from one of 30 technical micro-culture replicates from biological triplicates. The median and quartile of each distribution are indicated as black dashed and dotted lines, respectively. Numerical values are reported in Data C and D in S7 File.
(TIFF)

**S7 Fig. Principal component analysis of bacterial and phage transcriptomes.** (A) Principal component analysis of replicate transcriptomes of stationary phase B. thailandensis incubated for 4 h either in LB medium only (green squares), or in LB medium containing ciprofloxacin at 0.125× its MIC (blue diamonds), or LB medium containing phage at an MOI of 1 (black

circles), or in LB medium containing both ciprofloxacin at 0.125× its MIC and phage at an MOI of 1 (red triangles). (B) Corresponding principal component analysis of replicate transcriptomes of phage ΦBp-AMP1 after 4 h incubation with stationary phase B. thailandensis in LB medium only (open circles) or in LB medium containing ciprofloxacin at 0.125× its MIC (open triangles). Note the narrower x-axis range with respect to S6A Fig because of the largely overlapping phage transcriptomes. Numerical values are reported in S11 File.
(TIF)

**S8 Fig. Differential expression of genes involved in efflux, ATP generation, respiration, LPS transport and secretion during monotherapy or combination therapy with respect to untreated *B. thailandensis*.** Downregulated (negative Log2-fold change) or upregulated (positive Log2-fold change) genes of interest in stationary phase *B. thailandensis* after 4 h of monotherapy with ciprofloxacin at 0.125× MIC (blue empty bars) or combination therapy with ciprofloxacin at 0.125× MIC and phage at an MOI of 1 (red filled bars) with respect to untreated *B. thailandensis* incubated for 4 h in LB medium only. Corresponding numerical values are reported in S8 and S9 File.
(TIFF)

**S9 Fig. Single-cell morphology.** Distribution of single-cell lengths for (A) B. thailandensis incubated in LB medium only, (F) B. thailandensis incubated in LB medium containing phage at a concentration of $2\times10^8$ PFU ml$^{-1}$ and (K) B. thailandensis incubated in LB medium containing both phage at a concentration of $2\times10^8$ PFU ml$^{-1}$ and ciprofloxacin at 0.125× its MIC. Each distribution contains 500 single-cell length measurements carried out on bacteria hosted in different microfluidic compartments from different biological triplicate experiments over a period of 9 h exposure to each condition. Corresponding representative microscopy images are reported in (B-E), (G-J) and (L-O), respectively. Scale bar: 5 μm. Numerical values are reported in S17 File.
(TIF)

**S10 Fig. Plaque size in the presence of sub-inhibitory concentrations of antibiotics.** Distribution of sizes of plaques formed ΦBp-AMP1 when plated on B. thailandensis alone (black circles), or in the presence of ciprofloxacin (red squares), ampicillin (green triangles) or trimethoprim (purple diamonds) at 0.125× their respective MIC values. Each symbol reports a plaque size value measured on one of 150 plaques for each condition. Dashed and dotted horizontal lines represent the median and quartiles, respectively, of each distribution. **** indicates a p-value < 0.0001. Similar data were obtained when these antibiotics were used at 0.5× their respective MIC values. Numerical values are reported in S18 File.
(TIFF)

**S1 Table. Unique mutations identified in high- and low-resistance populations.** Mutant name, gene, gene product, annotation and unique mutation measured in the five high- and low-resistant populations indicated in Fig 1C and 1D.
(DOCX)

**S2 Table. Phage within uninfected *B. thailandensis* cultures.** Score of integrity for *Bulkholderia* phage ΦBp-AMP1, ΦE12-2, ΦE125, KL3 and vB_BmuP_KL4, and the *Ralstonia* phage RsoM1USA measured from 5 representative high-resistant mutant cultures, 5 representative low-resistant mutant cultures and 5 untreated control cultures, all in the absence of externally added phage. Detected phage sequences were given a score of integrity via PHASTER, with a maximum value of 150, scores above 90 were considered as intact prophages (blue shaded

tabs), scores below 70 were considered as incomplete (red shaded tabs).
(DOCX)

**S3 Table. Additivism between phage and antibiotics.** Antibiotic class, antibiotic molecule and range of antibiotic concentrations for which our model predicted additivism between phage and each antibiotic based on our experimental MIC values recorded for monotherapy and combination therapy using each antibiotic. Interactions between phage and antibiotics were considered to be additive if the probability that combination therapy was more effective in inhibiting bacterial growth than phage and antibiotic monotherapies was above 95%.
(DOCX)

**S1 File. Impact of phage ΦBp-AMP1 on the growth of *B. thailandensis*.** Data A: numerical values for the data reported in Fig 1A. Data B: numerical values for the data reported in Fig 1B. Data C: numerical values for the data reported in Figs 1C and S3.
(XLSX)

**S2 File. Impact of antibiotics on phage growth.** Numerical values for the data reported in S1 Fig.
(XLSX)

**S3 File. Heterogeneity in resistance to phage ΦBp-AMP1.** Numerical values for the data reported in S4 Fig.
(XLSX)

**S4 File. Impact of phage ΦBp-AMP1 on antibiotic inhibitory effect against *B. thailandensis*.** Data A-C: Numerical values for the data reported in Fig 2A–2C.
(XLSX)

**S5 File. Impact of antibiotic concentration on the interactions between antibiotics and phage ΦBp-AMP1.** Data A: Numerical values for the data reported in Fig 3. Data B: Numerical values for the data reported in S5 Fig.
(XLSX)

**S6 File. Impact of phage and antibiotic concentrations on the interactions between antibiotics and phage ΦBp-AMP1.** Numerical values for the data reported in Fig 4.
(XLSX)

**S7 File. Impact of phage ΦBp-AMP1 on antibiotic bactericidal effect against *B. thailandensis*.** Data A, B and E: Numerical values for the data reported in Fig 5. Data C and D: Numerical values for the data reported in S6 Fig.
(XLSX)

**S8 File. Differential expression of *B. thailandensis* genes after ciprofloxacin monotherapy relative to untreated control *B. thailandensis*.** Transcript base mean, log2 fold change, standard error of log2 fold change, p-value, adjusted p-value, gene identifier, gene and gene product for each of 4920 differentially expressed *B. thailandensis* genes, identified via Illumina sequencing, between ciprofloxacin monotherapy relative to untreated control *B. thailandensis*.
(CSV)

**S9 File. Differential expression of *B. thailandensis* genes after phage monotherapy relative to untreated control *B. thailandensis*.** Transcript base mean, log2 fold change, standard error of log2 fold change, p-value, adjusted p-value, gene identifier, gene and gene product for each of 5229 differentially expressed *B. thailandensis* genes, identified via Illumina sequencing,

between phage monotherapy relative to untreated control *B. thailandensis*.
(CSV)

**S10 File. Differential expression of *B. thailandensis* genes after combination therapy relative to untreated control *B. thailandensis*.** Transcript base mean, log2 fold change, standard error of log2 fold change, p-value, adjusted p-value, gene identifier, gene and gene product for each of 5215 differentially expressed *B. thailandensis* genes, identified via Illumina sequencing, between combination therapy relative to untreated control *B. thailandensis*.
(CSV)

**S11 File. Impact of phage ΦBp-AMP1 and ciprofloxacin on *B. thailandensis* transcriptome.** Numerical values for the data reported in S7 Fig.
(XLSX)

**S12 File. Gene ontology enrichment analysis of differentially expressed *B. thailandensis* genes between ciprofloxacin monotherapy and untreated control bacteria.** Identifier, description, number of genes contained, enrichment score, normalized enrichment score, p-value, adjusted p-value, q-value, rank, leading edge and identifier of each gene contained for each significantly enriched biological process, molecular function or cellular component in the comparison between ciprofloxacin monotherapy and untreated control bacteria.
(XLSX)

**S13 File. Gene ontology enrichment analysis of differentially expressed *B. thailandensis* genes between phage monotherapy and untreated control bacteria.** Identifier, description, number of genes contained, enrichment score, normalized enrichment score, p-value, adjusted p-value, q-value, rank, leading edge and identifier of each gene contained for each significantly enriched biological process, molecular function or cellular component in the comparison between phage monotherapy and untreated control bacteria.
(XLSX)

**S14 File. Gene ontology enrichment analysis of differentially expressed *B. thailandensis* genes between combination therapy and untreated control bacteria.** Identifier, description, number of genes contained, enrichment score, normalized enrichment score, p-value, adjusted p-value, q-value, rank, leading edge and identifier of each gene contained for each significantly enriched biological process, molecular function or cellular component in the comparison between combination therapy and untreated control bacteria.
(XLSX)

**S15 File. Differential expression of ΦBp-AMP1 genes after combination therapy relative to phage monotherapy.** Transcript base mean, log2 fold change, standard error of log2 fold change, p-value, adjusted p-value, gene identifier, gene and gene product for each of 41 differentially expressed ΦBp-AMP1 genes, identified via Illumina sequencing, between combination therapy relative to phage monotherapy.
(CSV)

**S16 File. Impact of phage ΦBp-AMP1 on the accumulation of ciprofloxacin in *B. thailandensis*.** Numerical values for the data reported in Fig 6.
(XLSX)

**S17 File. Impact of phage ΦBp-AMP1 on cell length of *B. thailandensis*.** Numerical values for the data reported in S9 Fig.
(XLSX)

**S18 File. Impact of antibiotics on the extent of plaques formed by phage ΦBp-AMP1.**
Numerical values for the data reported in S10 Fig.
(XLSX)

## Acknowledgments

We would like to thank the members of the MultiDefence Team for useful discussions and input.

## Author Contributions

**Conceptualization:** Stefano Pagliara.

**Data curation:** Samuel Kraus, Urszula Łapińska, Audrey Farbos, Stefano Pagliara.

**Formal analysis:** Samuel Kraus, Megan L. Fletcher, Urszula Łapińska, Evan Baker, Paul O'Neill, Aaron Jeffries, Edouard E. Galyov, Sunee Korbsrisate, Kay B. Barnes, Sarah V. Harding, Krasimira Tsaneva-Atanasova, Mark A T Blaskovich, Stefano Pagliara.

**Funding acquisition:** Aaron Jeffries, Sarah V. Harding, Krasimira Tsaneva-Atanasova, Mark A T Blaskovich, Stefano Pagliara.

**Investigation:** Samuel Kraus, Megan L. Fletcher, Urszula Łapińska, Krina Chawla, Evan Baker, Erin L. Attrill, Audrey Farbos.

**Methodology:** Samuel Kraus, Megan L. Fletcher, Urszula Łapińska, Krina Chawla, Evan Baker, Erin L. Attrill.

**Project administration:** Stefano Pagliara.

**Resources:** Stefano Pagliara.

**Software:** Evan Baker, Paul O'Neill.

**Supervision:** Mark A T Blaskovich, Stefano Pagliara.

**Visualization:** Samuel Kraus, Urszula Łapińska, Stefano Pagliara.

**Writing – original draft:** Samuel Kraus, Stefano Pagliara.

**Writing – review & editing:** Samuel Kraus, Megan L. Fletcher, Urszula Łapińska, Krina Chawla, Evan Baker, Erin L. Attrill, Paul O'Neill, Audrey Farbos, Aaron Jeffries, Edouard E. Galyov, Sunee Korbsrisate, Kay B. Barnes, Sarah V. Harding, Krasimira Tsaneva-Atanasova, Mark A T Blaskovich.

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
