## [Decision Letter · Decision Letter 0]

25 Feb 2024

Dear Dr. Pagliara,

Thank you very much for submitting your manuscript "Phage-induced efflux down-regulation boosts antibiotic efficacy" for consideration at PLOS Pathogens. As with all papers reviewed by the journal, your manuscript was reviewed by members of the editorial board and by several independent reviewers. In light of the reviews (below this email), we would like to invite the resubmission of a significantly-revised version that takes into account the reviewers' comments.

We recommend, as Reviewer 2 suggests, that you test the mechanism for ciprofloxacin-phage synergy by assessing dependence on the presence of the efflux system (i.e. examining mutant strains) and/or by showing a generalized effect of phage infection on efflux of other substrates (e.g. Nile Red). Further, you should provide data demonstrating improved sterilization of Burkholderia in animal models or in biofilm models. Last, we recommend that you provide data on phage resistance of the sequenced clones rather than their parent populations.

We cannot make any decision about publication until we have seen the revised manuscript and your response to the reviewers' comments. Your revised manuscript is also likely to be sent to reviewers for further evaluation.

Sincerely,

Gregory P Priebe, M.D.

Academic Editor

PLOS Pathogens

Nina Salama

Section Editor

PLOS Pathogens

Michael Malim

Editor-in-Chief

PLOS Pathogens

orcid.org/0000-0002-7699-2064

We recommend, as Reviewer 2 suggests, that you test the mechanism for ciprofloxacin-phage synergy by assessing dependence on the presence of the efflux system (i.e. examining mutant strains) and/or by showing a generalized effect of phage infection on efflux of other substrates (e.g. Nile Red). Further, you should provide data demonstrating improved sterilization of Burkholderia in animal models or in biofilm models. Last, we recommend that you provide data on phage resistance of the sequenced clones rather than their parent populations.

Reviewer's Responses to Questions

**Part I - Summary**

Reviewer #1: This manuscript describes experiments with phage, antibiotics and Burkholderia bacteria to look at synergies between phages and antimicrobials, and whether certain combinations would increase versus decrease efficacy of controlling bacterial growth. This was a very nice study, which I found to be convincing in the experiments and controls that were well-designed to test the stated hypotheses and to draw logical conclusions. This is a rare circumstance (in my experience) where a manuscript is so well-written and the study so nicely designed that I have no major concerns with the work. Nevertheless, I listed some minor concerns that the authors might consider when revising their paper.

Reviewer #2: The MS of Kraus et al. describes the effects of combination phage plus antibiotic therapy on stationary phase Burkholderia thailandensis. The authors demonstrate that a conditionally lytic phage (phiBp-AMP1) can decrease the MIC and minimum bactericidal concentration for a subset of antibiotics, although the effect is not observed for all classes of antibiotics (and is antagonistic for TMP/SMX).

The authors then carry out a transcriptomic (RNAseq) analysis of cultures ±phage/±ciprofloxacin and conclude that the synergy between phage and cipro is due to phage downregulation of an efflux system and “energy generation” systems. This is supported by observation of decreased ciprofloxacin-NBD fluorescence signal within cells concurrently exposed to phage as compared to those without phage infection (although the viability of the cipro-NBD plus phage cells isn't tested at the high (32 ug/ml) concentration of cipro-NBD used in this experiment).

Lastly, the mechanism of synergy between phage and beta-lactams (observed over a very narrow concentration window) or of the basis for antagonism between phage and TMP/SMX (is this a general effect of THFR inhibitors on phage replication?) is not examined in the study.

Reviewer #3: Interest in phage therapy has grown rapidly over the last few years and is becoming more widely accepted. However, application of phage therapy in many scenarios would be as an addition to existing antimicrobial treatments, rather than a direct replacement, in effect, creating a combination therapy. Therefore, a better understanding of the potential for synergy and/or antagonism between phages and antibiotics is urgently needed to enable rational design of phage-antibiotic co-therapies.

This study provides an interesting example of phage-antibiotic interactions and highlights a number of factors which are important for exploring these relationships within other systems. They explore the interaction between a Burkholderia phage, phiBp-AMP1, with multiple different antibiotics covering a range of clinically relevant antibiotic classes and modes of action. Further, by exploring the relationship between antibiotic concentration and phage multiplicity of infection, the authors demonstrate the variability in phage-antibiotic relationships: synergistic and/or antagonistic interactions may be observed only within a window of antibiotic concentrations, and outside this either antibiotic or phage treatments dominate the response. Promisingly, synergy between phage phiBp-AMP1 and both ciprofloxacin and ampicillin promote bacterial killing at sub-inhibitory antibiotic concentrations with low titres of phage required.

Finally, the authors were able to propose a convincing mechanism of action for the specific synergy between ciprofloxacin and phiBp-AMP1. Following up on a hypothesis informed by the effect of phage on bacterial transcriptomes which suggested down-regulation of an efflux pump in the presence of phage, using microfluidics the authors were able to demonstrate that phage phiBp-AMP1 promotes intracellular accumulation of ciprofloxacin.

Overall, this paper is a comprehensive exploration of how phage phiBp-AMP1 influences antibiotic efficacy in Burkholderia thaliandensis and provides valuable insight into the mechanisms by which phage-antibiotic synergy may play out. These results are likely to be of broad interest in the context of phage therapy. The experimental protocols, replication levels and statistical analysis all appear rigorous and well formed.

**Part II – Major Issues: Key Experiments Required for Acceptance**

Reviewer #1: No major issues.

Reviewer #2: The proposed mechanism for cipro-phage synergy is not further tested by showing a dependence of “phage synergy” on the presence of the efflux system (i.e. examining mutant strain), directly measuring bacterial cell membrane potential ± phage infection (using a phage-R mutant as a control) or by showing a generalized effect of phage infection on efflux of other substrates (eg Nile Red). The authors state that the combination of phage and antibiotics could be advantageous in situations where it is difficult to achieve high-antibiotic concentrations (in vivo, within biofilms), but do not provide data demonstrating improved sterilization of Burkholderia in animal models or in biofilm models.

Reviewer #3: none

**Part III – Minor Issues: Editorial and Data Presentation Modifications**

Reviewer #1: 1. The study did an excellent job of examining different phage MOIs and antibiotic concentrations. In development of therapies that leverage phage-antibiotic synergy, sometimes there is consideration of the relative timing of administering these antimicrobials. In particular, they could be co-administered, versus a different strategy involving either the phage or antimicrobial administered first and then ‘chased’ with the other at a later time point. Understandably, these types of manipulations fall outside the scope of the current study, and I am not asking that the authors conduct additional experiments. However, I would enjoy hearing their thoughts on the newly discovered biological mechanism in this study, and whether/how such differing strategies for administering the antimicrobials could matter.

2. This study uses B. thailandensis strain E264 as a model, which can be highly useful because studies can leverage the prior literature on model strains to more straightforwardly understand empirical, genetic etc. results. Although I favor such approaches, as I read the paper I became curious whether strain E264 is closely versus distantly related to clinically relevant bacteria. More broadly, I wonder whether the observations in this study would likely benefit development of therapeutic strategies against other Burkholderia bacteria species. Again, I believe such studies fall outside of the current work, and there is some time spent in the Discussion nicely describing prior work in these other pathogen systems. Nevertheless, I felt the authors might want to place the work in context of likely versus unlikely immediate usefulness in building out development strategies targeting many different B. thailandensis strains and/or other Burkholderia species. I am not asking the authors to be overly speculative, but perhaps they can address this briefly in the Discussion.

3. Related to the above comment, I felt the paper could better inform the reader of the health impact of Burkholderia bacteria, such as the patient communities particularly affected and/or the geographic regions that might be experiencing relatively high caseloads. The authors included some nice references, but they could include a bit more information in the text of the Introduction, for example.

Reviewer #2: 1. line 45 – define “Bptm”

2. line 51 – “as well as changes in the global environment” – not sure what this means here

3. line 59/61 – antibodies mentioned twice.

4. line 62 – “antimicrobial agents cannot change or adapt in real time” – this is also true of standard antimicrobials…and is it desirable for an antimicrobial agent to adapt in real time?

5. line 70 – “seven” years or “several” years?

6. line 77/78 – would be appropriate here to also discuss other ways in which phage/antibiotics synergize, e.g. downregulation of efflux systems that serve as phage receptors (as in work of Paul Turner and colleagues)

7. line 143 – “bacterial micro-cultures” – not clear if this is somehow different from small scale cultures in 96 well plates (methods don’t describe anything but this) – is it critical to have a certain inoculum size to observe the “zero” growth phenotype?

8. line 171 – what does it mean to be a ‘representative’ high- or low-resistant survivor? What is meant by “unique” mutations – they were observed as the only genetic change present in the genome as compared to the parent?

9. line 312 – this should be noted as “data not shown”

10. line 380 – authors note the emergence of a phage-resistant subpopulation that grows to high density; what implications does this have for using phage plus antibiotics in clinical practice?

11. line 452 – cipro-NBD is reportedly used at 32 ug/ml – what is the MIC for this modified cipro molecule? Are the bacteria viable in the combo phage/antibiotic condition?

12. line 472 (and abstract) – phage-antibiotic interactions are “underpinned by phage-induced downregulation…”. Underpinned is an imprecise word – the relationship between the phenotype and the proposed mechanism, however, is not established.

13. Fig 6 – what are the insets in panel C representing?

14. line 538 – “The outcome of phage-antibiotic therapy is often contradictory” – rephrase, this is not clear in its meaning.

Reviewer #3: L163-167 Was there variation in the growth of ‘survivor populations’ between ‘high’ and ‘low’ resistance in this follow-up assay in liquid media (i.e., OD) and/or in colony size?

L170-179. This is a relatively small sample size to infer how strength of resistance corelates to specific resistance mutations, especially considering that all mutations appear unique. Do you have data on the phage resistance of these individual sequenced clones, rather than their parent populations? E.g., growth in the presence/absence of phages, colony size to infer fitness costs.

L356. End of sentence missing?

L435. Would be interesting to see these specific examples included in Figure 6 to help compare the degree of enrichment/downregulation in context to other affected functions.

Figure 6A. I found it confusing to have different y-axis labels for plots A and B when they are aligned. It would be clearer to include all categories in both plots to enable a more informed comparison.

PLOS authors have the option to publish the peer review history of their article (what does this mean?). If published, this will include your full peer review and any attached files.

Reviewer #1: No

Reviewer #2: No

Reviewer #3: No

Figure File

---

## [Decision Letter · Decision Letter 1]

17 Jun 2024

Dear Dr. Pagliara,

Thank you very much for submitting your manuscript "Phage-induced efflux down-regulation boosts antibiotic efficacy" for consideration at PLOS Pathogens. As with all papers reviewed by the journal, your manuscript was reviewed by members of the editorial board and by several independent reviewers. The reviewers appreciated the attention to an important topic. Based on the reviews, we are likely to accept this manuscript for publication, providing that you modify the manuscript according to the review recommendations.

We ask that you focus on tempering your conclusions regarding the mechanism underlying phage-antibiotic synergy as suggested by Reviewer 2.

Sincerely,

Gregory P Priebe, M.D.

Academic Editor

PLOS Pathogens

David Skurnik

Section Editor

PLOS Pathogens

Michael Malim

Editor-in-Chief

PLOS Pathogens

orcid.org/0000-0002-7699-2064

We ask that you temper your conclusions regarding the mechanism underlying phage-antibiotic synergy.

Reviewer Comments (if any, and for reference):

Reviewer's Responses to Questions

**Part I - Summary**

Reviewer #1: (No Response)

Reviewer #2: The revised manuscript describes synergy between phage therapy and antimicrobial therapy against B. thailandensis, a difficult to treat bacterial pathogen.

Reviewer #3: The authors have thoroughly addressed my previous comments through new experiments and inclusion of new data figures. I have no further concerns.

**Part II – Major Issues: Key Experiments Required for Acceptance**

Reviewer #1: (No Response)

Reviewer #2: The authors have responded to reviewer comments with revised text and new figures/data. I appreciate their responses, but would continue to advise that they temper their conclusions regarding the mechanism underlying phage-antibiotic synergy as being increased antibiotic efflux. The cipro-NBD and Syto9 experiments that they show do not distinguish between increased permeabilization of the small molecule versus efflux (both effects can contribute to increased fluorescence given the experimental design) - and since their transcriptomic data document decreased expression of LPS transport genes as well as BpeEF-OprC this caveat should be acknowledged.

Reviewer #3: (No Response)

**Part III – Minor Issues: Editorial and Data Presentation Modifications**

Reviewer #1: (No Response)

Reviewer #2: (No Response)

Reviewer #3: (No Response)

PLOS authors have the option to publish the peer review history of their article (what does this mean?). If published, this will include your full peer review and any attached files.

Reviewer #1: No

Reviewer #2: No

Reviewer #3: No

Figure Files:

Data Requirements:

Please note that, as a condition of publication, PLOS' data policy requires that you make available all data used to draw the conclusions outlined in your manuscript. Data must be deposited in an appropriate repository, included within the body of the manuscript, or uploaded as supporting information. This includes all numerical values that were used to generate graphs, histograms etc.. For an example see here: http://www.plosbiology.org/article/info:doi%2F10.1371%2Fjournal.pbio.1001908#s5.

Reproducibility:

References:

---

## [Editor Report · Decision Letter 2]

21 Jun 2024

Dear Dr. Pagliara,

We are pleased to inform you that your manuscript 'Phage-induced efflux down-regulation boosts antibiotic efficacy' has been provisionally accepted for publication in PLOS Pathogens.

Best regards,

Gregory P Priebe, M.D.

Academic Editor

PLOS Pathogens

David Skurnik

Section Editor

PLOS Pathogens

Michael Malim

Editor-in-Chief

PLOS Pathogens

orcid.org/0000-0002-7699-2064
---

## [Editor Report · Acceptance letter]

24 Jun 2024

Dear Dr. Pagliara,

We are delighted to inform you that your manuscript, "Phage-induced efflux down-regulation boosts antibiotic efficacy," has been formally accepted for publication in PLOS Pathogens.

Best regards,

Michael Malim

Editor-in-Chief

PLOS Pathogens

orcid.org/0000-0002-7699-2064